# Multi-View Causal Representation Learning with Partial Observability

**Dingling Yao**[1,2], **Danru Xu**[3], **Sébastien Lachapelle**[7,8], **Sara Magliacane**[3,6], **Perouz Taslakian**[9], **Georg Martius**[4], **Julius von Kügelgen**[2,5], and **Francesco Locatello**[1]

[1]Institute of Science and Technology Austria   [2]Max Planck Institute for Intelligent Systems, Tübingen   [3]University of Amsterdam   [4]University of Tübingen   [5]University of Cambridge
[6]MIT-IBM Watson AI Lab   [7]Samsung - SAIT AI Lab   [8]Mila, Université de Montréal
[9]ServiceNow Research

## Abstract

We present a unified framework for studying the identifiability of representations learned from simultaneously observed views, such as different data modalities. We allow a *partially observed* setting in which each view constitutes a nonlinear mixture of a subset of underlying latent variables, which can be causally related. We prove that the information shared across all subsets of any number of views can be learned up to a smooth bijection using contrastive learning and a single encoder per view. We also provide graphical criteria indicating which latent variables can be identified through a simple set of rules, which we refer to as *identifiability algebra*. Our general framework and theoretical results unify and extend several previous works on multi-view nonlinear ICA, disentanglement, and causal representation learning. We experimentally validate our claims on numerical, image, and multi-modal data sets. Further, we demonstrate that the performance of prior methods is recovered in different special cases of our setup. Overall, we find that access to multiple partial views enables us to identify a more fine-grained representation, under the generally milder assumption of partial observability.

## 1 Introduction

Discovering latent structure underlying data has been important across many scientific disciplines, spanning neuroscience (Vigário et al., 1997; Brown et al., 2001), communication theory (Ristaniemi, 1999; Donoho, 2006), natural sciences (Wunsch, 1996; Chadan & Sabatier, 2012; Trapnell et al., 2014), and countless more. The underlying assumption is that many natural phenomena measured by instruments have a simple structure that is lost in raw measurements. In the famous cocktail party problem (Cherry, 1953), multiple speakers talk concurrently, and while we can easily record their overlapping voices, we are interested in understanding what individual people are saying. From the methodological perspective, such inverse problems became common in machine learning with breakthroughs in linear (Comon, 1994; Darmois, 1951; Hyvärinen & Erkki, 2000) and non-linear (Hyvarinen et al., 2019) Independent Component Analysis (ICA), and developed into deep learning methods for disentanglement (Bengio et al., 2013; Higgins et al., 2017). More recently, approaches to causal representation learning (Schölkopf et al., 2021) began relaxing the key assumption of independent latents central to prior work (the *independent* in ICA), allowing for and discovering (some) hidden causal relations (Brehmer et al., 2022; Lippe et al., 2022; Lachapelle et al., 2022; Zhang et al., 2023; Ahuja et al., 2023; Varici et al., 2023; Squires et al., 2023; von Kügelgen et al., 2023).

This problem is often modeled as a two-stage sampling procedure, where latent variables $\mathbf{z}$ are sampled i.i.d. from a distribution $p_{\mathbf{z}}$, and the observations $\mathbf{x}$ are functions thereof. Intuitively, the latent variables describe the causal structure underlying a specific environment, and they are only observed through sensor measurements, entangling them via so-called "*mixing functions*". Unfortunately, if these mixing functions are non-linear, the recovery of the latent variables is generally impossible, even if the latent variables are independent (Locatello et al., 2019; Hyvärinen & Pajunen, 1999). Following these negative results, the community has turned to settings that relax the i.i.d. condition in different ways. One particularly successful paradigm has been the assumption that data

is not independently sampled, and in fact, multiple observations may refer to the same realization of the latent variables. This multi-view setup has generated a flurry of results in ICA (Gresele et al., 2020; Zimmermann et al., 2021; Pandeva & Forré, 2023), disentanglement (Locatello et al., 2020; Klindt et al., 2021; Fumero et al., 2023; Lachapelle et al., 2023; Ahuja et al., 2022), and causal representation learning (von Kügelgen et al., 2021; Daunhawer et al., 2023; Brehmer et al., 2022).

This paper provides a unified framework for several identifiability results in observational multi-view causal representation learning under partial observability. We assume that different views need not be functions of *all* the latent variables, but only of some of them. For example, a person may undertake different medical exams, each shedding light on some of their overall health status (assumed constant throughout the measurements) but none offering a comprehensive view. An X-ray may show a broken bone, an MRI how the fracture affected nearby tissues, and a blood sample may inform about ongoing infections. Our framework also allows for an arbitrary number of views, each measuring partially overlapping latent variables. It includes multi-view ICA and disentanglement as special cases.

More technically, we prove that any shared information across arbitrary subsets of views and modalities can be learned up to a smooth bijection using contrastive learning. Non-shared information can also be identified if it is independent of other latent variables. With a single identifiability proof, our result implies the identifiability of several prior works in causal representation learning (von Kügelgen et al., 2021; Daunhawer et al., 2023), non-linear ICA (Gresele et al., 2020), and disentangled representations (Locatello et al., 2020; Ahuja et al., 2022). In addition to weaker assumptions, our framework retains the algorithmic simplicity of prior contrastive multi-view (von Kügelgen et al., 2021) and multimodal (Daunhawer et al., 2023) causal representation learning approaches. Allowing partial observability and arbitrarily many views, our framework is significantly more flexible than prior work, allowing us to identify shared information between all subsets of views and not just their joint intersection.

We highlight the following contributions:

1. We provide a unified framework for identifiability in observational multi-view causal representation learning with partial observability. This generalizes the multi-view setting in two ways: allowing (i) *any arbitrary number of views*, and (ii) partial observability with non-linear mixing functions. We prove that any shared information across arbitrary subsets of views and modalities can be learned up to a smooth bijection using contrastive learning and provide straightforward graphical criteria to categorize which latents can be recovered.

2. With a single proof, our result implies the identifiability of several prior works in causal representation learning, non-linear ICA, and disentangled representations as special cases.

3. We conduct experiments for various unsupervised and supervised tasks and empirically show that (i) the performance of prior works can be recovered using a special setup of our framework and (ii) our method indicates promising disentanglement capabilities with encoder-only networks.

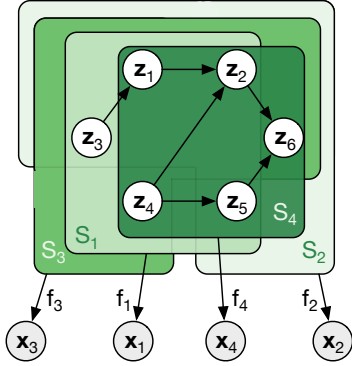

Figure 1: **Multi-View Setting with Partial Observability**, for Example 2.1 with $K=4$ views and $N=6$ latents. Each view $\mathbf{x}_k$ is generated by a subset $\mathbf{z}_{S_k}$ of the latent variables through a view-specific mixing function $f_k$. Directed arrows between latents indicate causal relations.

## 2 PROBLEM FORMULATION

We formalize the data generating process as a latent variable model. Let $\mathbf{z} = (\mathbf{z}_1, ..., \mathbf{z}_N) \sim p_{\mathbf{z}}$ be possibly dependent (causally related) latent variables taking values in $\mathcal{Z} = \mathcal{Z}_1 \times ... \times \mathcal{Z}_N$, where $\mathcal{Z} \subseteq \mathbb{R}^N$ is an open, simply connected latent space with associated probability density $p_{\mathbf{z}}$. Instead of directly observing $\mathbf{z}$, we observe a set of entangled measurements or views $\mathbf{x} := (\mathbf{x}_1, ..., \mathbf{x}_K)$. Importantly, we assume that each observed view $\mathbf{x}_k$ may only depend on some of the latent variables, which we call *"view-specific latents"* $\mathbf{z}_{S_k}$, indexed by subsets $S_1, ..., S_K \subseteq [N] = \{1, ..., N\}$. For any $A \subseteq [N]$, the subset of latent variables $\mathbf{z}_A$ and corresponding latent sub-space $\mathcal{Z}_A$ are given by:

$$\mathbf{z}_A := \{\mathbf{z}_j : j \in A\}, \qquad \mathcal{Z}_A := \bigtimes_{j \in A} \mathcal{Z}_j.$$

Similarly, for any $V \subseteq [K]$, the subset of views $\mathbf{x}_V$ and corresponding observation space $\mathcal{X}_V$ are:

$$\mathbf{x}_V := \{\mathbf{x}_k : k \in V\}, \qquad \mathcal{X}_V := \bigtimes_{k \in V} \mathcal{X}_k.$$

The *view-specific mixing functions* $\{f_k : \mathcal{Z}_{S_k} \to \mathcal{X}_k\}_{k \in [K]}$ are smooth, invertible mappings from the view-specific latent subspaces $\mathcal{Z}_{S_k}$ to observation spaces $\mathcal{X}_k \subseteq \mathbb{R}^{\dim(\mathbf{x}_k)}$ with $\mathbf{x}_k := f_k(\mathbf{z}_{S_k})$. Formally, the generative process for the views $\{\mathbf{x}_1, \ldots, \mathbf{x}_K\}$ is given by:

$$\mathbf{z} \sim p_{\mathbf{z}}, \qquad \mathbf{x}_k := f_k(\mathbf{z}_{S_k}) \quad \forall k \in [K],$$

i.e., each view $\mathbf{x}_k$ depends on latents $\mathbf{z}_{S_k}$ through a *mixing function* $f_k$, as illustrated in Fig. 1.

**Assumption 2.1** (General Assumptions). For the latent generative model defined above:

   (i) Each view-specific mixing function $f_k$ is a diffeomorphism;

  (ii) $p_{\mathbf{z}}$ is a smooth and continuous density on $\mathcal{Z}$ with $p_{\mathbf{z}} > 0$ almost everywhere.

**Example 2.1.** Throughout, we illustrate key concepts and results using the following example with $K=4$ views, $N=6$ latents, and dependencies among the $\mathbf{z}_k$ shown as a graphical model in Fig. 1.

$$\begin{aligned}
\mathbf{x}_1 &= f_1(\mathbf{z}_1, \mathbf{z}_2, \mathbf{z}_3, \mathbf{z}_4, \mathbf{z}_5), & \mathbf{x}_2 &= f_2(\mathbf{z}_1, \mathbf{z}_2, \mathbf{z}_3, \mathbf{z}_5, \mathbf{z}_6), \\
\mathbf{x}_3 &= f_3(\mathbf{z}_1, \mathbf{z}_2, \mathbf{z}_3, \mathbf{z}_4, \mathbf{z}_6), & \mathbf{x}_4 &= f_4(\mathbf{z}_1, \mathbf{z}_2, \mathbf{z}_4, \mathbf{z}_5, \mathbf{z}_6).
\end{aligned} \tag{2.1}$$

Consider a set $\mathbf{x}_V$ of jointly observed views, and let $\mathcal{V} := \{V_i \subseteq V : |V_i| \geq 2\}$ be the set of subsets $V_i \in \mathcal{V}$ indexing two or more views. For any subset of views $V_i$, we refer to the set of *shared* latent variables (i.e., those influencing each view in the set) as the "*content*" or "*content block*" of $V_i$. Formally, content variables $\mathbf{z}_{C_i}$ are obtained as intersections of view-specific indexing sets:

$$C_i = \bigcap_{k \in V_i} S_k \,. \tag{2.2}$$

Similarly, for each view $k \in V$, we can define the non-shared ("style") variables as $\mathbf{z}_{S_k \setminus C_i}$. We use $C$ and $\mathbf{z}_C$ without subscript to refer to the joint content across all observed views $\mathbf{x}_V$.

For Example 2.1, the content of $\mathbf{x}_{V_1} = (\mathbf{x}_1, \mathbf{x}_2)$ is $\mathbf{z}_{C_1} = (\mathbf{z}_1, \mathbf{z}_2, \mathbf{z}_3, \mathbf{z}_5)$; the content for all four views $\mathbf{x} = (\mathbf{x}_1, \mathbf{x}_2, \mathbf{x}_3, \mathbf{x}_4)$ jointly is $\mathbf{z}_C = (\mathbf{z}_1, \mathbf{z}_2)$, and the style for $\mathbf{x}_1$ is $\mathbf{z}_{S_1 \setminus C} = (\mathbf{z}_3, \mathbf{z}_4, \mathbf{z}_5)$.

*Remark* 2.2 ("Content-Style" Terminology). We adopt these terms from von Kügelgen et al. (2021), but note that, in our setting, they are relative to a specific subset of views. Unlike in some prior works (Gresele et al., 2020; von Kügelgen et al., 2021; Daunhawer et al., 2023), style variables are generally not considered irrelevant, but may also be of interest and can sometimes be identified by other means (e.g., from other subsets of views or because they are independent of content).

Our goal is to show that we can simultaneously identify multiple content blocks given a set of jointly observed views under weak assumptions. This extends previous work (Gresele et al., 2020; von Kügelgen et al., 2021; Daunhawer et al., 2023) where only one block of content variables is considered. Isolating the shared content blocks from the rest of the view-specific style information, the learned representation (estimated content) can be used in downstream pipelines, such as classification tasks (Lachapelle et al., 2023; Fumero et al., 2023). In the best case, if each latent component is represented as one individual content block, we can learn a fully disentangled representation (Higgins et al., 2018; Locatello et al., 2020; Ahuja et al., 2022). To this end, we restate the definition of *block-identifiability* (von Kügelgen et al., 2021, Defn 4.1) for the multi-modal, multi-view setting:

**Definition 2.3** (Block-Identifiability). The true content variables $\mathbf{c}$ are *block-identified* by a function $g : \mathcal{X} \to \mathbb{R}^{\dim(\mathbf{c})}$ if the inferred content partition $\hat{\mathbf{c}} = g(\mathbf{x})$ contains *all and only* information about $\mathbf{c}$, i.e., if there exists some smooth *invertible* mapping $h : \mathbb{R}^{\dim(\mathbf{c})} \to \mathbb{R}^{\dim(\mathbf{c})}$ s.t. $\hat{\mathbf{c}} = h(\mathbf{c})$.

Note that the inferred content variables $\hat{c}$ can be a set of *entangled* latent variables rather than a single one. This differentiates our paper from the line of work on *disentanglement* (Locatello et al., 2020; Fumero et al., 2023; Lachapelle et al., 2023), which seek to disentangle *individual* latent factors and can thus be considered as special cases of our framework with content block sizes equal to one.

## 3    IDENTIFIABILITY THEORY

**High-Level Overview.** This section presents a unified framework for studying identifiability from multiple partial views: we start by establishing identifiability of the shared content block $\mathbf{z}_C$ from *any number of partially observed views* (Thm. 3.2). The downside of this approach is that if we seek to

learn content from different subsets, we need to train an *exponential* number of encoders for the same modality, one for each subset of views. We, therefore, extend this result and show that by considering any subset of the jointly observed views, various blocks of content variables can be identified by *one single* view-specific encoder (Thm. 3.8). After recovering multiple content blocks *simultaneously*, we show in Cors. 3.9 to 3.11 that a qualitative description of the data generative process such as in Fig. 1 can be sufficient to determine exactly the extent to which individual latents or groups thereof can be identified and disentangled. Full proofs are included in App. C.

**Definition 3.1** (Content Encoders). Assume that the content size $|C|$ is given for any jointly observed views $\mathbf{x}_V$. The content encoders $G := \{g_k : \mathcal{X}_k \to (0, 1)^{|C|}\}_{k \in V}$ consist of smooth functions mapping from the respective observation spaces to the $|C|$-dimensional unit cube.

**Theorem 3.2** (Identifiability from a *Set* of Views). *Consider a set of views $\mathbf{x}_V$ satisfying Asm. 2.1, and let $G$ be a set of content encoders (Defn. 3.1) that minimizes the following objective*

$$\mathcal{L}(G) = \underbrace{\sum_{k < k' \in V} \mathbb{E}\left[\|g_k(\mathbf{x}_k) - g_{k'}(\mathbf{x}_{k'})\|_2\right]}_{\text{Content alignment}} - \underbrace{\sum_{k \in V} H\left(g_k(\mathbf{x}_k)\right)}_{\text{Entropy regularization}}, \qquad (3.1)$$

*where the expectation is taken w.r.t. $p(\mathbf{x}_V)$ and $H(\cdot)$ denotes differential entropy. Then the shared content variable $\mathbf{z}_C := \{\mathbf{z}_j : j \in C\}$ is block-identified (Defn. 2.3) by $g_k \in G$ for any $k \in V$.*

> **Intuition.** The *alignment* enforces the content encoders $g_k$ only to encode content and discard styles, while the maximized *entropy* implies uniformity and thus invertibility. For Example 2.1, recall that the joint content is $C = \cap_{k \in [4]}\{S_k\} = \{1, 2\}$. Thm. 3.2 then states that, for each $k = 1, 2, 3, 4$, the content encoders $G = \{g_k : \mathcal{X}_k \to (0, 1)^{|C|}\}$ which minimize the loss in eq. (3.1) are actually invertible mappings of the ground truth content $\{\mathbf{z}_1, \mathbf{z}_2\}$, i.e., $g_k(\mathbf{x}_k) = h_k(\mathbf{z}_1, \mathbf{z}_2)$ for smooth invertible functions $h_k : \mathcal{Z}_1 \times \mathcal{Z}_2 \to (0, 1)^2$.

**Discussion.** Thm. 3.2 provides a learning algorithm to infer *one* jointly shared content block for *all* observed views in a set, extending prior results that only consider two views (von Kügelgen et al., 2021; Daunhawer et al., 2023; Locatello et al., 2020). However, to discover another content block $C_i$ w.r.t. a subset of views $V_i \subset V$ as defined in § 2, we need to train another set of encoders, since the dimensionality of the content might change. Ideally, we would like to learn *one view-specific encoder $r_k$* that maps from the observation space $\mathcal{X}_k$ to some $|S_k|$-dimensional manifold and can block-identify all shared contents $\mathbf{z}_{C_i}$ using *one* training run, combined with separate *content selectors*.

**Definition 3.3** (View-Specific Encoders). The *view-specific encoders* $R := \{r_k : \mathcal{X}_k \to \mathcal{Z}_{S_k}\}_{k \in V}$ consist of smooth functions mapping from the respective observation spaces to the view-specific latent space, where the dimension of the $k^{\text{th}}$ latent space $|S_k|$ is assumed known for all $k \in V$.

> **Intuition.** The view-specific encoders learn *all view-related content blocks simultaneously*, instead of training a combinatorial number of networks (as would be implied by Thm. 3.2). The view-specific encoders should learn not only a single block of content variables, but instead learn to recover *all* shared latents in a way that makes it *easy to extract various different content blocks using simple readout functions*. This is possible by construction, e.g., if each $r_k$ learns to invert the ground truth mixing $f_k$. Inspired by this idea, we introduce *content selectors*.

**Definition 3.4** (Selection). A selection $\oslash$ operates between two vectors $a \in \{0, 1\}^d, b \in \mathbb{R}^d$ s.t.

$$a \oslash b := [b_j : a_j = 1, j \in [d]]$$

**Definition 3.5** (Content Selectors). The content selectors $\Phi := \{\phi^{(i,k)}\}_{V_i \in \mathcal{V}, k \in V_i}$ with $\phi^{(i,k)} \in \{0, 1\}^{|S_k|}$ perform selection (Defn. 3.4) on the encoded information: for any subset $V_i$ and view $k \in V_i$ we have the selected representation: $\phi^{(i,k)} \oslash \hat{\mathbf{z}}_{S_k} = \phi^{(i,k)} \oslash r_k(\mathbf{x}_k)$, with $\left\|\phi^{(i,k)}\right\|_0 = \left\|\phi^{(i,k')}\right\|_0$ for all $V_i \in \mathcal{V}, k, k' \in V_i$.

> **Intuition.** Using *learnable* binary weights $\Phi$, the content selectors $\phi^{(i,k)}$ should pick out those latents among the representation $\hat{\mathbf{z}}_{S_k}$ extracted by $r_k$ that belong to the content block $C_i$ shared among $V_i$. For Example 2.1, consider a learned representation $r_1(\mathbf{x}_1) = (\hat{\mathbf{z}}_1, \hat{\mathbf{z}}_2, \hat{\mathbf{z}}_3, \hat{\mathbf{z}}_4, \hat{\mathbf{z}}_5)$. Applying a content selector with weight $\phi^{(i=1,k=1)} = [1, 1, 1, 0, 1]$ then yields: $(\hat{\mathbf{z}}_1, \hat{\mathbf{z}}_2, \hat{\mathbf{z}}_3, \hat{\mathbf{z}}_5)$.

**What is missing?** While aligning various content blocks based on the same representation $r_k(\mathbf{x}_k)$ should promote *disentanglement*, maximizing the entropy $H(r_k(\mathbf{x}_k))$ of the learned representation (as in Thm. 3.2) promotes *uniformity*. The latter implies invertibility of the encoders (Zimmermann et al., 2021), which is necessary for block-identifiability (Defn. 2.3). However, since a uniform representation has independent components by definition, disentanglement and uniformity cannot be achieved simultaneously unless all ground truth latents are mutually independent (a strong assumption we are not willing to make). Thus, to *theoretically* achieve invertibility while preserving disentanglement, we introduce a set of auxiliary *projection* functions.

**Definition 3.6** (Projections). The set of projections $T := \{t_k\}_{k \in V}$ consist of functions $t_k : \mathcal{Z}_{S_k} \to (0,1)^{|S_k|}$ mapping each view-specific latent space to a hyper unit-cube of the same dimension $|S_k|$.

> **Intuition.** The projection functions can be understood as mathematical tools: by maximizing the entropy and thus enforcing uniformity of *projected representations* $t_k \circ r_k(\mathbf{x}_k)$, we can show that $r_k$ needs to be invertible without interfering with the disentanglement of different content blocks.

**What if the content dimension is unknown?** In Thm. 3.2 we assumed that the size $|C|$ of the shared content block is known, and the encoders map to a space of dimension $|C|$. In the following, we do *not* assume that the content size is given. Instead, we will show that the correct content block can still be discovered by ensuring that as much information as possible is shared across any given subset of views. To this end, we define the following information-sharing regularizer.

**Definition 3.7** (Information-Sharing Regularizer). The following regularizer penalizes the $L_0$-norm $\|\cdot\|_0$ of the content selectors $\Phi$: $\mathrm{Reg}(\Phi) := -\sum_{V_i \in \mathcal{V}} \sum_{k \in V_i} \|\phi^{(i,k)}\|_0$ .

> **Intuition.** $\mathrm{Reg}(\Phi)$ sums the number of shared latents over $V_i \subseteq \mathcal{V}$ and $k \in V_i$. It decreases when $\phi^{(i,k)}$ contains more ones, i.e., more latents are shared across views $k \in V_i$. Thus, $\mathrm{Reg}(\Phi)$ encourages the encoders to reuse the learned latents and maximize the shared information content.

**Theorem 3.8** (View-Specific Encoder for Identifiability). *Let $R, \Phi, T$ respectively be any view-specific encoders (Defn. 3.3), content selectors (Defn. 3.1) and projections (Defn. 3.6) that solve the following constrained optimization problem:*

$$\min \mathrm{Reg}(\Phi) \qquad \textit{subject to:} \qquad R, \Phi, T \in \arg\min \mathcal{L}(R, \Phi, T) \tag{3.2}$$

*where*

$$\mathcal{L}(R, \Phi, T) = \sum_{V_i \in \mathcal{V}} \sum_{\substack{k,k' \in V_i \\ k < k'}} \mathbb{E}\underbrace{\left[\left\|\phi^{(i,k)} \oslash r_k(\mathbf{x}_k) - \phi^{(i,k')} \oslash r_{k'}(\mathbf{x}_{k'})\right\|_2\right]}_{\textit{Content alignment}} - \sum_{k \in V} \underbrace{H(t_k \circ r_k(\mathbf{x}_k))}_{\textit{Entropy}},$$
$$\tag{3.3}$$

*Then for any subset of views $V_i \in \mathcal{V}$ and any view $k \in V_i$, $\phi^{(i,k)} \oslash r_k$ block-identifies (Defn. 2.3) the shared **content** variables $\mathbf{z}_{C_i}$, as defined in eq. (2.2).*

> **Intuition.** For Example 2.1, the representation for $\mathbf{x}_1$ obtained by minimizing eq. (3.2) is given by $\hat{\mathbf{z}} := r_1(\mathbf{x}_1) = r_1(f_1(\mathbf{z}_{S_1})) = r_1 \circ f_1(\mathbf{z}_1, \mathbf{z}_2, \mathbf{z}_3, \mathbf{z}_4, \mathbf{z}_5)$. Consider the following two subsets of views $V_1, V_2 \in \mathcal{V}$ containing $\mathbf{x}_1$, but sharing different content blocks $C_1, C_2$:
>
> $$\mathbf{x}_{V_1} = \{\mathbf{x}_1, \mathbf{x}_2\}, \quad \mathbf{z}_{C_1} = \{\mathbf{z}_1, \mathbf{z}_2, \mathbf{z}_3, \mathbf{z}_5\}, \quad \mathbf{x}_{V_2} = \{\mathbf{x}_1, \mathbf{x}_3\}, \quad \mathbf{z}_{C_2} = \{\mathbf{z}_1, \mathbf{z}_2, \mathbf{z}_3, \mathbf{z}_4\}.$$
>
> Then one of the optimal solutions of the selectors learned by Thm. 3.8 could be
>
> $$\phi^{(i=1,k=1)} = [1,1,1,0,1], \qquad \phi^{(i=2,k=1)} = [1,1,1,1,0].$$
>
> Hence, the composed results of the selectors and the view-specific encoder $r_1$ give:
>
> $$\phi^{(i=1,k=1)} \oslash \hat{\mathbf{z}} = h_{i=1,k=1}(\mathbf{z}_1, \mathbf{z}_2, \mathbf{z}_3, \mathbf{z}_5), \qquad \phi^{(i=2,k=1)} \oslash \hat{\mathbf{z}} = h_{i=2,k=1}(\mathbf{z}_1, \mathbf{z}_2, \mathbf{z}_3, \mathbf{z}_4)$$
>
> where $h_{i,k=1}$ is some smooth bijection, for both $i = 1, 2$.

**Discussion.** Note that Equation (3.2) can be rewritten as a regularized loss $\mathcal{L}_{\mathrm{Reg}}(R, \Phi, T) = \mathcal{L}(R, \Phi, T) + \alpha \cdot \mathrm{Reg}(\Phi)$ with a sufficiently small regularization coefficient $\alpha \geq 0$. Overall, Thm. 3.8 further weakens the assumptions of Thm. 3.2 in that no content size is required. However,

minimizing the information-sharing regularizer is highly non-convex, and having only a finite number of samples makes finding the global optimum challenging. In practice, we could use *Gumbel Softmax* (Jang et al., 2016) for unsupervised learning, and consider content sizes as hyper-parameters or follow the approach by Fumero et al. (2023) for supervised classification tasks. Empirically, we will see that some of the requirements that are needed in theory can be realistically dropped, and different approximations are possible, e.g., incorporating problem-specific knowledge.

After discovering various content blocks, we are further interested in how to infer more information from the learned content blocks. For example, can we identify $\mathbf{z}_{C_3} := \{\mathbf{z}_1, \mathbf{z}_2, \mathbf{z}_3\} = \mathbf{z}_{C_1 \cap C_2}$? This perspective motivates our next results, which focus on how to infer new information based on the previously identified blocks: ***Identifiability Algebra***.

Let $\mathbf{z}_{C_1}, \mathbf{z}_{C_2}$ with $C_1, C_2 \subseteq [N]$ be two identified blocks of latents. Then it holds for $C_1, C_2$ that:

**Corollary 3.9** (Identifiability Algebra: Intersection). *The intersection $\mathbf{z}_{C_1 \cap C_2}$ can be block-identified.*

**Corollary 3.10** (Identifiability Algebra: Complement). *If $C_1 \cap C_2$ is independent of $C_1 \backslash C_2$, then the complement $\mathbf{z}_{C_1 \backslash C_2}$ can be block-identified.*

**Corollary 3.11** (Identifiability Algebra: Union). *If $C_1 \cap C_2$, $C_1 \backslash C_2$ and $C_2 \backslash C_1$ are mutually independent, then the union $\mathbf{z}_{C_1 \cup C_2}$ can be block-identified.*

**Discussion.** While Cor. 3.9 refines the identified block of information into smaller intersections, Cors. 3.10 and 3.11 allows to extract "style" variables as defined w.r.t. some specific views, under the assumption that they are independent of the content block, as discussed by Lyu et al. (2021). However, our setup is more general, as we can not only explain the independent style variables between pairs of *observations*, but also between *learned content representations*. Thus, by iteratively applying Cor. 3.10 we can generalize the statement to any number of identified content blocks. Combining Cors. 3.9 to 3.11 we can immediately tell which part can be block-identified from a set of views $\mathcal{V}$, given a graphical model representation such as Fig. 1 and subject to technical assumptions underlying our main results. Applying Cors. 3.9 to 3.11 iteratively on identified blocks can possibly *disentangle* each individual factors of variation, providing a novel approach for disentanglement. If all variables can be isolated up to element-wise nonlinear transformations, we can learn the causal relations by assuming the original link are nonlinear with additive noise. This exactly corresponds to a post-nonlinear model, whose graphic structure can be further identified using causal discovery algorithms (Zhang & Chan, 2006; Zhang & Hyvärinen, 2009).

## 4 RELATED WORK AND SPECIAL CASES OF OUR THEORY

Our framework unifies several prior work, including *multi-view nonlinear ICA* (Gresele et al., 2020), *weakly-supervised disentanglement* (Locatello et al., 2020; Ahuja et al., 2022) and *content-style identification* (von Kügelgen et al., 2021; Daunhawer et al., 2023). Tab. 1 shows a summarized (non-exhaustive) list of related works and their respective graphical models that can be considered as special cases. The graphical setups of the individual works can be recovered from our framework (Fig. 1) by varying the number of observed views and causal relations.

In addition, we present a short overview of other related work which can be connected with our theoretical results, including *causal representation learning* (Sturma et al., 2023; Silva et al., 2006; Adams et al., 2021; Kivva et al., 2021; Cai et al., 2019; Xie et al., 2020; 2022; Morioka & Hyvärinen, 2023; Morioka & Hyvarinen, 2023), *mutual information-based contrastive learning* (Tian et al., 2020; Tsai et al., 2020; Tosh et al., 2021), *latent correlation maximization* (Andrew et al., 2013; Benton et al., 2017; Lyu & Fu, 2020; Lyu et al., 2021), *nonlinear ICA without auxiliary variables* (Willetts & Paige, 2021) and *multitask disentanglement with sparse classifiers* (Lachapelle et al., 2023; Fumero et al., 2023). Further discussion is given in App. B. We remark that several approaches here consider the setting where two observations are generated through an intervention on some latent variable(s). This is sometimes written in the graphical model as two nodes connected by an arrow (shown in the graphs in Tab. 1 as dashed lines $--\rightarrow$) indicating the pre- and post-intervention versions of the same variable(s). We stress that this does *not* constitute an example of partial observability. In our setting, latent variables can be simply unobserved, regardless of whether or not they were subject to an intervention.

**Causal representation learning.** In the context of causal representation learning (CRL), Sturma et al. (2023) also explicitly consider partial observability in a *linear, multi-domain* setting. Several other works on *linear* CRL from *i.i.d.* data could also be viewed as assuming partial observability, since

Table 1: A non-exhaustive summary of *special cases* of our theory and their graphical models. An asterisk (*) indicates works that have view-specific latents that are not of interest for identifiability.

| Method | Graph | Dependent Latents | Multi-Modal | Partial Observability | > 2 Views |
|---|---|---|---|---|---|
| Schölkopf et al. (2016) | | ✗ | ✓ | ✓ | ✗ |
| Gresele et al. (2020) | | ✗ | ✓ | ✗* | ✓ |
| Locatello et al. (2020) | | ✗ | ✗ | ✗ | ✗ |
| Ahuja et al. (2022) | | ✓ | ✗ | ✗ | ✓ |
| von Kügelgen et al. (2021) | | ✓ | ✗ | ✗ | ✗ |
| Daunhawer et al. (2023) | | ✓ | ✓ | ✗* | ✗ |
| Ours | Fig. 1 | ✓ | ✓ | ✓ | ✓ |

they often rely on graphical conditions which enforce each measured variable to depend on a single (a "pure" child) or only a few latents (Silva et al., 2006; Adams et al., 2021; Kivva et al., 2021; Cai et al., 2019; Xie et al., 2020; 2022). In our framework, each view $\mathbf{x}_k$ instead constitutes a *nonlinear* mixture of *several* latents. Merging partially observed causal structure has been studied without a representation learning component by Gresele et al. (2022); Mejia et al. (2022); Guo et al. (2023).

**Mutual Information-based Contrastive Learning.** Tian et al. (2020); Tsai et al. (2020); Tosh et al. (2021) empirically showcase the success of contrastive learning in extracting task-related information across multiple views, if the augmented views are redundant to the original data regarding task-related information (Tian et al., 2020). From this point of view, the *redundant* task-information can be interpreted as shared content between the views, for which our theory (Thms. 3.2 and 3.8) may provide theoretical explanations for the improved performance in downstream tasks.

**Latent Correlation Maximization.** Prior work (Andrew et al., 2013; Benton et al., 2017; Lyu & Fu, 2020; Lyu et al., 2021) showed that maximizing the correlation between the learned representation is equivalent to our *content alignment* principle (eq. (3.1)). The additional invertibility constraint on the learned encoder in their setting is enforced by entropy regularization (eq. (3.1)), as explained by Zimmermann et al. (2021). However, their theory is limited to pairs of views and full observability, while we generalize it to any number of partially observed views.

**Nonlinear ICA without Auxiliary Variables.** Willetts & Paige (2021) shows nonlinear ICA problem can be solved using non-observable, learnable, clustering task variables $u$, to replace the observed auxiliary variable in other nonlinear ICA approaches (Hyvarinen et al., 2019). While we explicitly require the learned representation to be aligned in a continuous space within the content block, Willetts & Paige (2021) impose a *soft* alignment constraint to encourage the encoded information to be similar within a cluster. In practice, the soft alignment requirement can be easily coded in our framework by relaxing the *content alignment* with an equivalence class in terms of cluster membership.

**Multi-task Disentanglement with Sparse Classifiers.** Our setup is slightly different from that of Lachapelle et al. (2023); Fumero et al. (2023) as they focus on multiple classification tasks using shared encoding and sparse linear readouts. Their sparse classifier head jointly enforces the sufficient representation (regarding the specific classification task, while we aim for the invertibility of the encoders) and a soft alignment up to a linear equivalence class (relaxing our hard alignment). However, the identifiability principles we use are similar: sufficient representation (entropy regularization), alignment and information sharing. While our results can be easily extended to allow for alignment up to a linear equivalence class, their identifiability theory crucially only covers independent latents.

## 5 EXPERIMENTS

First, we validate Thms. 3.2 and 3.8 using numerical simulations in a *fully controlled* synthetic setting. Next, we conduct experiments on visual (and text) data demonstrating different special cases that are unified by our theoretical framework (§ 4) and how we extend them. We use InfoNCE (Oord et al., 2018) and BarlowTwins (Zbontar et al., 2021) to estimate eqs. (3.1) and (3.2). The *content alignment* is computed by the numerator (positive pairs) in InfoNCE and the *entropy regularization* is estimated by the denominator (negative pairs). Further remarks on contrastive learning and entropy regularization are in App. E. For the evaluation, we follow a standard evaluation protocol (von Kügelgen et al., 2021) and predict the ground truth latents from the learned representation $g_k(\mathbf{x}_k)$, using kernel ridge regression for *continuous* latent variables, and logistic regression for *discrete* ones, respectively. Then, we report the *coefficient of determination $R^2$* to show the correlation between the learned and ground truth latent variables. An $R^2$ close to one between the learned and ground truth variables means that the learned variables are modelling correctly the ground truth, indicating block-identifiability (Defn. 2.3). However, $R^2$ is limited as a metric, since any style variable that strongly depends on a content variable would also become predictable, thus showing a high $R^2$ score.

### 5.1 NUMERICAL EXPERIMENT: THEORY VALIDATION

**Experimental Setup.** We generate synthetic data following eq. (2.1). The latent variables are sampled from a Gaussian distribution $\mathbf{z} \sim \mathcal{N}(0, \Sigma_{\mathbf{z}})$, where possible *causal* dependencies can be encoded through $\Sigma_{\mathbf{z}}$. The *view-specific mixing functions $f_k$* are implemented by randomly initialized invertible MLPs for each view $k \in \{1, \ldots 4\}$. We report here the $R^2$ scores for the case of *independent* variables, because it is easier to interpret than the $R^2$ scores in the *causally dependent* case, for which we show that the learned representation still contains *all and only* the content information in App. D.1.

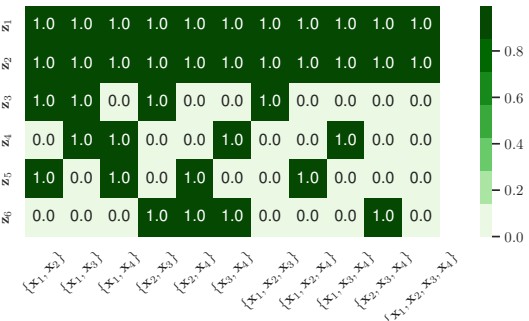

Figure 2: **Theory Validation**: Average $R^2$ across multiple views generated from *independent* latents.

**Discussion.** Fig. 2 shows how the averaged $R^2$ changes when including more views, with the y-axis denoting the ground truth latents and the x-axis showing learned representation from different subsets of views. As shown in Example 2.1 and Fig. 1, the content variables are *consistently* identified, having $R^2 \approx 1$, while the *independent* style variables are non-predictable ($R^2 \approx 0$). This numerical result shows that the learned representation explains almost all variation in the content block but nothing from the independent styles, which validates Thms. 3.2 and 3.8.

### 5.2 SELF-SUPERVISED DISENTANGLEMENT

**Experimental Setup.** We compare our method (Thm. 3.8) with Ada-GVAE (Locatello et al., 2019), on *MPI-3D complex* (Gondal et al., 2019) and *3DIdent* (Zimmermann et al., 2021) image datasets. We did not compare with Ahuja et al. (2022), since their method needs to know which latent is perturbed, even when guessing the offset. We experiment on a pair of views $(\mathbf{x}_1, \mathbf{x}_2)$ where the second view $\mathbf{x}_2$ is obtained by randomly perturbing a subset of latent factors of $\mathbf{x}_1$, following (Locatello et al., 2019). We provide more details about the datasets and the experiment setup in App. D.2. As shown in Tab. 2, our method outperformed the autoencoder-based Ada-GVAE (Locatello et al., 2020), using only an encoder and computing contrastive loss in the latent space.

Table 2: **Self-Supervised Disentanglement Performance Comparison** on *MPI-3D complex* (Gondal et al., 2019) and *3DIdent* (Zimmermann et al., 2021), between our method and Ada-GVAE (Locatello et al., 2020).

|  | DCI disentanglement ↑ | |
| --- | --- | --- |
|  | **MPI3D complex** | **3DIdent** |
| Ada-GVAE | 0.11± 0.008 | 0.09± 0.019 |
| Ours | **0.42**± 0.020 | **0.30**± 0.04 |

**Discussion.** As both methods are theoretically identifiable, we hypothesize that the improvement comes from avoiding reconstructing the image, which is more difficult on visually complex data. This hypothesis is supported by the fact that self-supervised contrastive learning has far exceeded the performance of autoencoder-based representation learning in both classification tasks (Chen et al., 2020; Caron et al., 2021; Oquab et al., 2023) and object discovery (Seitzer et al., 2022).

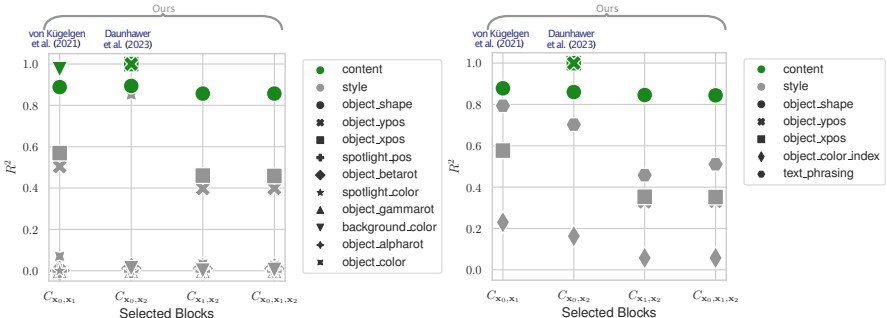

Figure 3: **Simultaneous Multi-Content Identification using View-Specific Encoders.** Experimental results on *Multimodal3DIdent*. *Left*: Image latents (averaged between two image views) *Right*: Text latents.

### 5.3 MULTI-MODAL CONTENT-STYLE IDENTIFIABILITY UNDER PARTIAL OBSERVABILITY

**Experimental setup.** We experiment on a set of *three* views $(\text{img}_0, \text{img}_1, \text{txt}_0)$ extending both (Daunhawer et al., 2023; von Kügelgen et al., 2021), which are limited to two views, either two images or one image and its caption. The second image view $\text{img}_1$ is generated by perturbing a subset of latents of $\text{img}_0$ as in (von Kügelgen et al., 2021). Notice that this setup provides perfect partial observability, because the text is generated using text-specific modality variables that are not involved in any image views e.g., *text phrasing*. We train *view-specific encoders* to learn *all content blocks simultaneously* and predict individual latent variables from the each *learned* content blocks. We assume access to the ground truth content indices to better match the baselines, but we relax this in App. D.4.

**Discussion.** Fig. 3 reports the $R^2$ on the ground truth latent values, predicted from the *simultaneously learned multiple content blocks* $(C_{\mathbf{x}_0,\mathbf{x}_1}, C_{\mathbf{x}_0,\mathbf{x}_2}, C_{\mathbf{x}_1,\mathbf{x}_2}, C_{\mathbf{x}_0,\mathbf{x}_1,\mathbf{x}_2}$, respectively). We remark that this *single* experiment recovers both experimental setups from (von Kügelgen et al., 2021, Sec 5.2), (Daunhawer et al., 2023, Sec 5.2): $C_{\mathbf{x}_0,\mathbf{x}_1}$ represents the content block from the image pairs $(\text{img}_0, \text{img}_1)$, which aligns with the setting in (von Kügelgen et al., 2021) and $C_{\mathbf{x}_0,\mathbf{x}_2}$ shows the content block from the multi-modal pair $(\text{img}_0, \text{txt}_0)$, which is studied by Daunhawer et al. (2023). We observe that the same performance for both prior works (von Kügelgen et al., 2021; Daunhawer et al., 2023) has been successfully reproduced from our single training process, which verifies the effectiveness and efficiency of Thm. 3.8. Extended evaluation and more experimental details are provided in App. D.4.

### 5.4 MULTI-TASK DISENTANGLEMENT

**Experimental setup** We follow Example 2.1 with *latent causal relations* to verify that: (i) the improved classification performance from (Lachapelle et al., 2023; Fumero et al., 2023) originates from the fact that the task-related information is shared across multiple views (different observations from the same class) and (ii) this information can be identified (Thm. 3.2), *even though the latent variables are not independent*. This explains the good performance of (Fumero et al., 2023) on real-world data sets, where the latent variables are likely not independent, violating their theory.

**Discussion.** We synthetically generate the labels by linear/nonlinear labeling functions on the shared content values to resemble (Lachapelle et al., 2023; Fumero et al., 2023). As expected, the learned representation significantly eases the classification task and achieves an accuracy of 0.99 with linear and nonlinear labeling functions within 1k update steps, even with latent causal relations. This experimental result justifies that the success in the empirical evaluation of (Fumero et al., 2023) can be explained by our theoretical framework, as discussed in § 4.

### 6 DISCUSSION AND CONCLUSION

This paper revisits the problem of identifying possibly dependent latent variables under multiple partial non-linear measurements. Our theoretical results extend to an arbitrary number of views, each potentially measuring a strict subset of the latent variables. In our experiments, we validate our claims and demonstrate how prior work can be obtained as a special case of our setting. While our assumptions are relatively mild, we still have notable gaps between theory and practice, thoroughly discussed in App. E. In particular, we highlight discrete variables and finite-sample errors as common gaps, which we address only empirically. Interestingly, our work offers potential connections with work in the causality literature (Triantafillou et al., 2010; Gresele et al., 2022; Mejia et al., 2022; Guo et al., 2023). Discovering hidden causal structures from overlapping but not simultaneously observed marginals (e.g., via views collected in different experimental studies at different times) remains open for future works.

REPRODUCIBILITY STATEMENT

The datasets used in (§ 5) are published by Gondal et al. (2019); Zimmermann et al. (2021); von Kügelgen et al. (2021); Daunhawer et al. (2023). Results provided in the experiments section (§ 5) can be reproduced using the implementation details provided in App. D. The code is available at https://github.com/CausalLearningAI/multiview-crl. The part of implementation to replicate the experiments of Fumero et al. (2023) in § 5.4 was kindly provided by the authors upon request, and we do not include it in the git repository.

ACKNOWLEDGEMENTS

This work was initiated at the Second Bellairs Workshop on Causality held at the Bellairs Research Institute, January 6–13, 2022; we thank all workshop participants for providing a stimulating research environment. Further, we thank Cian Eastwood, Luigi Gresele, Stefano Soatto, Marco Bagatella and A. René Geist for helpful discussion. GM is a member of the Machine Learning Cluster of Excellence, EXC number 2064/1 – Project number 390727645. JvK and GM acknowledge support from the German Federal Ministry of Education and Research (BMBF) through the Tübingen AI Center (FKZ: 01IS18039B). The research of DX and SM was supported by the Air Force Office of Scientific Research under award number FA8655-22-1-7155. Any opinions, findings, and conclusions or recommendations expressed in this material are those of the author(s) and do not necessarily reflect the views of the United States Air Force. We also thank SURF for the support in using the Dutch National Supercomputer Snellius. SL was supported by an IVADO excellence PhD scholarship and by Samsung Electronics Co., Ldt. DY was supported by an Amazon fellowship, the International Max Planck Research School for Intelligent Systems (IMPRS-IS) and the ISTA graduate school. Work done outside of Amazon.

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

# Appendix

## Table of Contents

## A    NOTATION AND TERMINOLOGY

$C$          Index set for shared content variables

$\mathbf{z}_C$          Shared content variables

$\mathcal{V}$          Collection of subset of views from $V$

$V_i$          Index set for subset of views in set of views $V$

$C_i$          Index set for shared content variables from subset of views $V_i \in \mathcal{V}$

$\mathbf{z}_{C_i}$          Shared content variables from subset of views $V_i$

$N$          Number of latents

$K$          Number of views

$j$          Index for latent variables

$k$          Index for views

$\mathcal{Z}$          Latent space

$\mathcal{X}_k$          Observational space for $k$-th view

$\mathbf{x}_k$          Observed $k$-th view

$S_k$          Index set for $k$-th view-specific latents

$V$          $\{1, \ldots, l\}$

## B    RELATED WORK AND SPECIAL CASES OF OUR THEORY

We present our identifiability results from § 3 as a unified framework implying several prior works in multi-view nonlinear ICA, disentanglement, and causal representation learning.

**Multi-View Nonlinear ICA**   Gresele et al. (2020) extend the idea of nonlinear ICA introduced by Hyvarinen et al. (2019, Sec. 3) by allowing a more flexible relationship between the latents and the auxiliary variables: instead of imposing *conditional independence* the shared source information $\mathbf{c}$ on some auxiliary observed variables, Gresele et al. (2020) *associate* the shared information source with some view-specific noise variable $n_i$ through some smooth mapping $g_k$:

$$\mathbf{x}_k = f_k(g_k(\mathbf{c}, \mathbf{n}_k)), \qquad k \in [K].$$

Define the composition of the view-specific function $f_k$ and the noise corruption function $g_k$ as a new mixing function $\tilde{f}_k := f_k \circ g_k.$, then each view $\mathbf{x}_k, k \in [K]$ is generated by a *view-specific mixing function* $\tilde{f}_k$ which takes the shared content $\mathbf{c}$ and some additional unobserved noise variable $n_k$ as input, that is: $\mathbf{x}_k = \tilde{f}_k(\mathbf{c}, \mathbf{n}_k)$. In this case, the shared source information $\mathbf{s}$ together with the view-specific latent noise $n_k$ defines the view-specific latents $S_k$ in our notation. The shared source $\mathbf{c}$ corresponds to our content variables. Our results Thm. 3.2 can be considered as a generalized version of Gresele et al. (2020, Theorem 8) for multiple views, by removing the additivity constraint of the corruption function $g$ (Gresele et al., 2020, Sec 3.3).

Willetts & Paige (2021); Kivva et al. (2022); Liu et al. (2022) show the nonlinear ICA problem can be solved using non-observable, learnable, clustering task variables $u$ to replace the observed the auxiliary variable in the traditional nonlinear ICA literature (Hyvärinen & Pajunen, 1999). Conditioning on the *same* latents on various clustering task enforces recovering the true latent factors (up to a bijective mapping). The idea of utilizing the clustering task goes hand in hand with contrastive self-supervised learning. Clustering itself can be considered as a soft relaxation of our hard global invariance condition in eq. (3.2), in the sense they enforce the shared, task-relevant features to be *similar* within a cluster but not necessarily having the exact same value.

**Weakly-Supervised Representation Learning**   Locatello et al. (2020) consider a pair of views (e.g., images) $(\mathbf{x}_1, \mathbf{x}_2)$ where $\mathbf{x}_2$ is obtained by perturbing a subset of the generating factors of $\mathbf{x}_1$. Formally,

$$\mathbf{x}_1 = f(\mathbf{z}) \qquad \mathbf{x}_2 = f(\tilde{\mathbf{z}}) \qquad \mathbf{z}, \tilde{\mathbf{z}} \in \mathbb{R}^d, \tag{B.1}$$

where $\mathbf{z}_S = \tilde{\mathbf{z}}_S$ while $\mathbf{z}_{\bar{S}} \neq \tilde{\mathbf{z}}_{\bar{S}}$ for some subset of latents $S \subseteq [d]$. In this case, $\mathbf{z}_S$ is the shared content between the pair of views $\mathbf{x}_1, \mathbf{x}_2$. According to the adaptive algorithm (Sec 4.1 Locatello et al., 2020), the shared content is computed by averaging the encoded representation from $\mathbf{x}_1, \mathbf{x}_2$ across the shared dimensions, that is:

$$g(\mathbf{x}_k)_j \leftarrow a(g(\mathbf{x}_1)_j, g(\mathbf{x}_2)_j) \qquad j \in S \tag{B.2}$$

By substituting the extracted representation using the averaged value, Locatello et al. (2020) achieve the same invariance condition as enforced in the first term of eq. (3.1). The amortized encoder $g$ is trained to maximize the ELBO (Kingma & Welling, 2013) over the pair of views, which is equivalent to minimizing the reconstruction loss

$$\mathbb{E}\left[\mathbf{x} - dec(g(\mathbf{x}_1))\right] + \mathbb{E}\left[\mathbf{x}_2 - dec(g(\mathbf{x}_2))\right]. \tag{B.3}$$

The reconstruction loss is minimized when the compression is lossless, or equivalently, the learned representation is uniformly distributed (Zimmermann et al., 2021). The uniformity of the representation, i.e., the lossless compression, can be enforced by maximizing the entropy term, as defined in eq. (3.1). Theoretically, Locatello et al. (2020, Theorem 1.) have shown that the shared content $\mathbf{z}_S$ can be recovered up to permutation, which aligns with our results Thm. 3.8 that the shared content can be inferred up to a smooth invertible mapping.

Ahuja et al. (2022) extend Locatello et al. (2020) by exploring more perturbations options to achieve full *disentanglement*. The main results (Ahuja et al., 2022, Theorem 8) state that each individual factor of a $d-$dimensional latents can be recovered (up to a bijection) when we augment the original observation with $d$ views, each obtained perturbing one unique latent component. This can be explained by Thm. 3.8 because any $(d-1)$ views from this set would share exactly one latent component, which makes it identifiable. Although the theoretical claim by Ahuja et al. (2022) is to some extent aligned with our theory, in practice, they explicitly require knowledge of the ground truth content indices while we do not necessarily.

**Mutual Information-based Framework**   Tian et al. (2020); Tsai et al. (2020) argue that the self-supervised signal should be approximately redundant to the task-related information. The self-supervised learning methods are based on extracting the task-relevant information (by maximizing the mutual information between the extracted representation $\hat{\mathbf{z}}_{\mathbf{x}}$ of the input $\mathbf{x}$ and the self-supervised signal $\mathbf{s}$: $I(\hat{\mathbf{z}}_{\mathbf{x}}, \mathbf{s})$) and discarding the task-irrelevant information conditioned on task $T$: $I(\mathbf{x}, \mathbf{s} \mid T)$. The mutual information $I(\hat{\mathbf{z}}_{\mathbf{x}}, \mathbf{s})$ is maximized if $\mathbf{s}$ is a deterministic function (for example, a MLP) of $\hat{\mathbf{z}}$ (Amjad & Geiger, 2020, Theorem 1.). Since the mutual information remains invariant under deterministic transformation of the random variables, we have:

$$\max I(\hat{\mathbf{z}}, s) = \max I(\hat{\mathbf{z}}, g(s)) = \max I(\hat{\mathbf{z}}, \hat{\mathbf{z}}) = \max H(\hat{\mathbf{z}}) \tag{B.4}$$

which is equivalent to maximizing the entropy of the learned representations, as given in eq. (3.1). Coupled with the empirically shown strong connection between the task-related information and shared content across multiple views (Tian et al., 2020; Tsai et al., 2020; Tosh et al., 2021; Lachapelle et al., 2023; Fumero et al., 2023), our results (Thms. 3.2 and 3.8) provides a theoretical explanation for these approaches. As the shared content between the original view and self-supervised signal is proven to be related to the ground truth task-related information through a smooth invertible function, it is reasonable to see the usefulness of this high quality representation in downstream tasks.

**Latent Correlation Maximization**   Similar alignment conditions, as given in eq. (C.1), have been widely studied in the latent correlation maximization / latent component matching literature (Andrew et al., 2013; Benton et al., 2017; Lyu & Fu, 2020; Lyu et al., 2021). Lyu et al. (2021, Theorem 1.) show that, by imposing additional invertibility constraint on the encoders latent correlation maximization across two views leads to identification of the shared component, up to an invertible function. This theoretical result can be considered as a explicit special case of Thm. 3.2, where we extend the identifiability proof to more than two multi-modal views.

**Content-Style Identification**   Our work is most closely related to (von Kügelgen et al., 2021; Daunhawer et al., 2023), while our results extended prior work that purely focused on identifiability from pair of views (von Kügelgen et al., 2021; Daunhawer et al., 2023). von Kügelgen et al. (2021, Theorem 4.2) presented a special case of Thm. 3.2 where the set size $l = 2$ and the mixing function $f_1 = f_2$ for both views; Daunhawer et al. (2023, Theorem 1) formulate another special case of Thm. 3.2 by allowing multi modality in the pair of views, but coming with the restriction that the view-specific modality variables have to be independent from others. From the data generating perspective, our work differs from prior work in the sense that all of the entangled views are simultaneously generated, each based on view-specific set of latent, while prior work generate the augmented (second) view by perturbing some style variables. In our case, "style" is relative to specific views. Style variables could become the content block for some set of views (Thm. 3.2) and thus be identifiable or can be inferred as independent complement block of the content (Cor. 3.10). Kong et al. (2022) have proven identifiability for *independent* partitions in the latent space but mostly focus on the domain adaptation tasks where additional targets are required as supervision signals.

**Multi-Task Disentanglement**   Lachapelle et al. (2023); Fumero et al. (2023); Zheng et al. (2022) differs from our theory in the sense that their sparse classifier head jointly enforces the lossless compression (which we do with the entropy regularization) and a soft alignment up to a linear transformation (relaxing our hard alignment). In their setting, the different views are images of the same class and their augmentations sampled from a given task and the selector variable is implemented with the linear classifier. The identifiability principles we use of lossless compression, alignment, and information-sharing are similar. With this, we can explain the result that task-related and task-irrelevant information can be disentangled as blocks, as given in Lachapelle et al. (2023, Theorem 3.1), Fumero et al. (2023, Proposition 1.). With our theory, their identifiability results extend to non-independent blocks, which is an important case that is not covered in the original works.

## C   PROOFS

### C.1   PROOF FOR THM. 3.2

Our proof follows the steps from von Kügelgen et al. (2021) with slight adaptation:

1. We show in Lemma C.1 that the lower bound of the loss eq. (3.1) is zero and construct encoders $\{g_k^* : \mathcal{X}_k \to (0, 1)^{|C|}\}_{k \in V}$ that reach this lower bound;

2. Next, we show in Lemma C.3 that for any set of encoders $\{g_k\}_{k \in V}$ that minimizes the loss, each learned $g_k(\mathbf{x}_k)$ depends only on the shared content variables $\mathbf{z}_C$, i.e. $g_k(\mathbf{x}_k) = h_k(\mathbf{z}_C)$ for some smooth function $h_k : \mathcal{Z}_C \to (0,1)^{|C|}$.

3. We conclude the proof by showing that every $h_k$ is invertible using Proposition 1 (Zimmermann et al., 2021, Proposition 5.).

We rephrase each step as a separate lemma and use them to complete the final proof for Thm. 3.2.

**Lemma C.1** (Existence of Optimal Encoders). *Consider a jointly observed set of views $\mathbf{x}_V$, satisfying Asm. 2.1. Let $S_k \subseteq [N]$, $k \in V$ be view-specific indexing sets of latent variables and define the shared coordinates $C := \bigcap_{k \in V} S_k$. For any content encoders $G := \{g_k : \mathcal{X}_k \to (0,1)^{|C|}\}_{k \in V}$ (Defn. 3.1), we define the following objective:*

$$\mathcal{L}(G) = \sum_{\substack{k,k' \in V \\ k < k'}} \mathbb{E}\left[\|g_k(\mathbf{x}_k) - g_{k'}(\mathbf{x}_{k'})\|_2\right] - \sum_{k \in V} H\left(g_k(\mathbf{x}_k)\right) \tag{C.1}$$

*where the expectation is taken with respect to $p(\mathbf{x}_V)$ and where $H(\cdot)$ denotes differential entropy. Then the global minimum of the loss (eq. (C.1)) is lower bounded by zero, and there exists a set of content encoders Defn. 3.1 which obtains this global minimum.*

*Proof.* Consider the objective function $\mathcal{L}(G)$ defined in eq. (C.1), the global minimum of $\mathcal{L}(G)$ is obtained when the first term (alignment) is minimized and the second term (entropy) is maximized. The alignment term is minimized to zero when $g_k$ are perfectly aligned for all $k \in V$, i.e., $g_k(\mathbf{x}_k) = g_{k'}(\mathbf{x}_{k'})$ for all $\mathbf{x}_V \sim p_{\mathbf{x}_V}$. The second term (entropy) is maximized to zero *only* when $g_k(\mathbf{x}_k)$ is uniformly distributed on $(0,1)^{|C|}$ for all views $k \in V$.

To show that there exists a set of smooth functions: $G := \{g_k\}_{k \in V}$ that minimizes $\mathcal{L}(G)$, we consider the inverse function of the ground truth mixing function $f_k^{-1}{}_{1:|C|}$, w.l.o.g. we assume that the content variables are at indices $1 : |C|$. This inverse function exists and is a smooth function given by Asm. 2.1(i) that each mixing function $f_k$ is a smooth invertible function. By definition, we have $f_k^{-1}{}_{1:|C|}(\mathbf{x}_k) = \mathbf{z}_C$ for $k \in V$.

Next, we define a function $\mathbf{d}$ using *Darmois construction* (Darmois, 1951) as follows:

$$d^j(\mathbf{z}_C) := F_j(z_j|\mathbf{z}_{1:j-1}) \qquad j \in \{1, \ldots, |C|\}, \tag{C.2}$$

where $F_j$ denotes the conditional cumulative distribution function (CDF) of $z_j$ given $\mathbf{z}_{1:j-1}$, i.e. $F_j(z_j|\mathbf{z}_{1:j-1}) := \mathbb{P}(Z_j \leq z_j|\mathbf{z}_{1:j-1})$. By construction, $\mathbf{d}(\mathbf{z}_C)$ is uniformly distributed on $(0,1)^{|C|}$. Moreover, $\mathbf{d}$ is smooth because $p_{\mathbf{z}}$ is a smooth density by Asm. 2.1(ii) and because conditional CDF of smooth densities is smooth

Finally, we define

$$g_k := \mathbf{d} \circ f_k^{-1}{}_{1:|C|} : \mathcal{X}_k \to (0,1)^{|C|}, \quad k \in V, \tag{C.3}$$

which is a smooth function as a composition of two smooth functions.

Next, we show that the function set $G$ as constructed above attains the global minimum of $\mathcal{L}(G)$. Given that $f_k^{-1}{}_{1:|C|}(\mathbf{x}_k) = f_{k'}^{-1}{}_{1:|C|}(\mathbf{x}_{k'}) = \mathbf{z}_C$, $\forall k, k' \in V$, we have:

$$\mathcal{L}(G) = \sum_{\substack{k,k' \in V \\ k < k'}} \mathbb{E}\left[\|g_k(\mathbf{x}_k) - g_{k'}(\mathbf{x}_{k'})\|_2\right] - \sum_{k \in V} H\left(g_k(\mathbf{x}_k)\right)$$

$$= \sum_{\substack{k,k' \in V \\ k < k'}} \mathbb{E}\left[\|\mathbf{d}(\mathbf{z}_C) - \mathbf{d}(\mathbf{z}_C)\|_2\right] - \sum_{k \in V} H\left(\mathbf{d}(\mathbf{z}_C)\right) \tag{C.4}$$

$$= 0,$$

where $\mathbf{z}_C$ is the shared content variables thus the first term (alignment) equals zero; and since $\mathbf{d}(\mathbf{z}_C)$ is uniformly distributed on $(0,1)^{|C|}$, the second term (entropy) is also zero.

To this end, we have shown that there exists a set of smooth encoders $G := \{g_k\}_{k \in V}$ with $g_k$ as defined in eq. (C.3) which minimizes the objective $\mathcal{L}(G)$ in eq. (C.1). $\qquad \square$

**Lemma C.2** (Conditions of Optimal Encoders). *Assume the same set of views* $\mathbf{x}_V$ *as introduced in Lemma C.1, then for any set of smooth encoders* $G := \{g_k : \mathcal{X}_k \to (0,1)^{|C|}\}_{k \in V}$ *to obtain the global minimum (zero) of the objective* $\mathcal{L}(G)$ *in eq. (C.1), the following two conditions have to be fulfilled:*

- **Invariance**: *All extracted representations* $\hat{\mathbf{z}}_k := g_k(\mathbf{x}_k)$ *must align across the views from the set $V$ almost surely:*

$$g_k(\mathbf{x}_k) = g_{k'}(\mathbf{x}_{k'}) \quad \forall k, k' \in V \quad a.s. \tag{C.5}$$

- **Uniformity**: *All extracted representations* $\hat{\mathbf{z}}_k := g_k(\mathbf{x}_k)$ *must be uniformly distributed over the hyper-cube* $(0,1)^{|C|}$.

*Proof.* Given that $G = \operatorname{argmin} \mathcal{L}(G)$, we have by Lemma C.1:

$$\mathcal{L}(G) = \sum_{k,k' \in V} \mathbb{E}\left[\|g_k(\mathbf{x}_k) - g_{k'}(\mathbf{x}_{k'})\|_2\right] - \sum_{k \in V} H(g_k(\mathbf{x}_k)) = 0 \tag{C.6}$$

The minimum $L(G) = 0$ leads to following conditions:

$$\mathbb{E}\left[\|g_k(\mathbf{x}_k) - g_{k'}(\mathbf{x}_{k'})\|_2\right] = 0 \quad \forall k, k' \in V, k < k' \tag{C.7}$$
$$H(g_k(\mathbf{x}_k)) = 0 \quad \forall k \in V \tag{C.8}$$

where eq. (C.7) indicates the invariance condition holds for all views $x_k$ and smooth encoders $g_k \in G$ almost surely; and eq. (C.8) implies that the encoded information $g_k(\mathbf{x}_k)$ must be uniformly distributed on $(0,1)^{|C|}$. $\square$

**Lemma C.3** (Content-Style Isolation from Set of Views). *Assume the same set of views* $\mathbf{x}_V$ *as introduced in Lemma C.1, then for any set of smooth encoders* $G := \{g_k : \mathcal{X}_k \to (0,1)^{|C|}\}_{k \in V}$ *that satisfies the* **Invariance** *condition (eq. (C.5)), the learned representation can only be dependent on the content variables* $\mathbf{z}_C := \{\mathbf{z}_j : j \in C\}$, *not any style variables* $\mathbf{z}_k^s := \mathbf{z}_{S_k \setminus C}$ *for all $k \in V$.*

*Proof.* Note that the learned representation can be rewritten as:

$$g_k(\mathbf{x}_k) = g_k(f_k(\mathbf{z}_{S_k})) \quad k \in V, \tag{C.9}$$

we define

$$h_k := g_k \circ f_k \quad k \in V. \tag{C.10}$$

Following the second step of the proof from von Kügelgen et al. (2021, Thm. 4.2), we show by contradiction that both $h_k(\mathbf{z}_{S_k})$ for all $k \in V$ can only depend on the shared content variables $\mathbf{z}_C$.

Let $k \in V$ be any view from the jointly observed set, suppose *for a contradiction* that $h_k^c := h_k(\mathbf{z}_{S_k})_{1:|C|}$ depends on some component $z_q$ from the view-specific latent variables $\mathbf{z}_k^s$:

$$\exists q \in \{1, \ldots, \dim(\mathbf{z}_k^s)\}, \mathbf{z}_{S_k} = (\mathbf{z}_C^*, \mathbf{z}_k^{s*}) \in \mathcal{Z}_k, \quad s.t. \quad \frac{\partial h_k^c}{\partial z_q}(\mathbf{z}_C^*, \mathbf{z}_k^{s*}) \neq 0, \tag{C.11}$$

which means that partial derivative of $h_k^c$ w.r.t. some latent variable $z_q \in \mathbf{z}_k^s$ is non-zero at some point $\mathbf{z}_{S_k} = (\mathbf{z}_C^*, \mathbf{z}_k^{s*}) \in \mathcal{Z}_k$. Since $h_k^c$ is smooth, its first-order (partial) derivatives are continuous. By continuity of the partial derivatives, $\frac{\partial h_1^c}{\partial z_q}$ must be non-zero in a neighborhood of $(\mathbf{z}_C^*, \mathbf{z}_k^{s*})$, i.e.,

$$\exists \eta > 0 \quad s.t. \quad z_q \to h_k^c(\mathbf{z}_C^*, \mathbf{z}_{k-q}^{s*}, z_q) \quad \text{is strictly monotonic on } (z_q - \eta, z_q + \eta), \tag{C.12}$$

where $\mathbf{z}_{k-q}^{s*}$ denotes the remaining view-specific style variables except $z_q$.

Next, we define an auxiliary function for each pair of views $(k, k')$ with $k, k' \in V, k < k'$: $\psi_{k,k'} : \mathcal{Z}_C \times \mathcal{Z}_{S_k \setminus C} \times \mathcal{Z}_{S_{k'} \setminus C} \to \mathbb{R}_{\geq 0}$

$$\begin{aligned} \psi_{k,k'}(\mathbf{z}_C, \mathbf{z}_k^s, \mathbf{z}_{k'}^s) :&= |h_k^c(\mathbf{z}_C, \mathbf{z}_k^s) - h_{k'}^c(\mathbf{z}_C, \mathbf{z}_{k'}^s)| \\ &= |h_k^c(\mathbf{z}_{S_{k'}}) - h_{k'}^c(\mathbf{z}_{S_{k'}})| \geq 0. \end{aligned} \tag{C.13}$$

Summarizing the pairwise auxiliary functions, we have $\psi : \mathcal{Z}_C \times \prod_{k \in V} \mathcal{Z}_{S_k \setminus C} \to \mathbb{R}_{\geq 0}$ as follows:

$$\psi(\mathbf{z}_C, \{\mathbf{z}_k^{\mathrm{s}}\}_{k \in V}) := \sum_{\substack{k,k' \in V \\ k < k'}} \left| h_k^{\mathrm{c}}(\mathbf{z}_C, \mathbf{z}_k^{\mathrm{s}}) - h_{k'}(\mathbf{z}_C, \mathbf{z}_{k'}^{\mathrm{s}}) \right|$$

$$= \sum_{\substack{k,k' \in V \\ k < k'}} \left| h_k^{\mathrm{c}}(\mathbf{z}_{S_{k'}}) - h_{k'}^{\mathrm{c}}(\mathbf{z}_{S_{k'}}) \right| \geq 0 \tag{C.14}$$

To obtain a contradiction to the invariance condition in Lemma C.2, it remains to show that $\psi$ from eq. (C.14) is *strictly positive* with a probability greater than zero w.r.t. the true generating process $p$; in other words, there has to exist at least one pair of views $(k, k')$ s.t. $\psi_{k,k'} > 0$ with a probability greater than zero regarding $p$.

Since $q \in S_k \setminus C$, there exists at least one view $k' \neq k$ s.t. $q \notin S_{k'}$ (otherwise the content block $C$ would contain $q$). We choose exactly such a pair of views $k, k'$.

Depending whether there is a zero point $z_q^0$ of $\psi$ within the region $(z_q - \eta, z_q + \eta)$, there are two cases to consider:

- If there is no zero-point $z_q^0 \in (z_q - \eta, z_q + \eta)$ s.t. $\psi_{k,k'}\left(\mathbf{z}_C^*, (\mathbf{z}_{k-q}^{\mathrm{s}*}, z_q^0), \mathbf{z}_{k'}^{\mathrm{s}*}\right) = 0$, then it implies

$$\psi_{k,k'}\left(\mathbf{z}_C^*, (\mathbf{z}_{k-q}^{\mathrm{s}*}, z_q), \mathbf{z}_{k'}^{\mathrm{s}*}\right) > 0 \quad \forall z_q \in (z_q - \eta, z_q + \eta). \tag{C.15}$$

  So there is an open set $A := (z_q - \eta, z_q + \eta) \subseteq \mathcal{Z}_q$ such that the equation $\psi$ in eq. (C.14) is strictly positive.

- Otherwise, there is a zero point $z_q^0$ from the interval $(z_q - \eta, z_q + \eta)$ with

$$\psi_{k,k'}\left(\mathbf{z}_C^*, (\mathbf{z}_{k-q}^{\mathrm{s}*}, z_q^0), \mathbf{z}_{k'}^{\mathrm{s}*}\right) = 0 \qquad z_q^0 \in (z_q - \eta, z_q + \eta), \tag{C.16}$$

  then strict monotonicity from eq. (C.12) implies that $\psi_{k,k'} > 0$ for all $z_q$ in the neighborhood of $z_q^0$, therefore:

$$\psi(\mathbf{z}_C, \{\mathbf{z}_k^{\mathrm{s}}\}_{k \in V}) > 0 \quad \forall z_q \in A := (z_q - \eta, z_q^0) \cup (z_q^0, z_q + \eta). \tag{C.17}$$

Since $\psi$ is a sum of compositions of two smooth functions (absolute different of two smooth functions), $\psi$ is also smooth. Consider the open set $\mathbb{R}_{>0}$ and note that, under a continuous function, pre-images of open sets are *always open*. For the continuous function $\psi$, its pre-image $\mathcal{U}$ corresponds to an *open set*:

$$\mathcal{U} \subseteq \mathcal{Z}_C \times \prod_{k \in V} \mathcal{Z}_{S_k \setminus C} \tag{C.18}$$

in the domain of $\psi$ on which $\psi$ is strictly positive. Moreover, since eq. (C.17) indicated that for all $z_q \in A$, the function $\psi$ is strictly positive, which means:

$$\{\mathbf{z}_C^*\} \times \prod_{k: q \in S_k \setminus C} \left(\{\mathbf{z}_{k-q}^{\mathrm{s}*}\} \times A\right) \times \prod_{k: q \notin S_k} \{\mathbf{z}_k^{\mathrm{s}*}\} \subseteq \mathcal{U}, \tag{C.19}$$

hence, $\mathcal{U}$ is *non-empty*.

Given by Asm. 2.1 (ii) that $p_{\mathbf{z}}$ is smooth and fully supported ($p_{\mathbf{z}} > 0$ almost everywhere), the non-empty set $\mathcal{U}$ is also fully supported by $p_{\mathbf{z}}$, which indicates:

$$\mathbb{P}\left(\psi(\mathbf{z}_C, \{\mathbf{z}_k^{\mathrm{s}}\}_{k \in V}) > 0\right) \geq \mathbb{P}(\mathcal{U}) > 0, \tag{C.20}$$

where $\mathbb{P}$ denotes the probability w.r.t. the true generative process $p$.

According to Lemma C.2, the invariance condition and uniformity conditions has to be fulfilled. To this end, we have shown that the assumption eq. (C.11) leads to an contradiction to the invariance condition eq. (C.5). Hence, assumption eq. (C.11) cannot hold, i.e., $h_k^{\mathrm{c}}$ does not depend on any view-specific style variable $z_q$ from $\mathbf{z}_k^{\mathrm{s}}$. It is only a function of the shared content variables $\mathbf{z}_C$, that is, $\hat{\mathbf{z}}_k^{\mathrm{c}} = h_k^{\mathrm{c}}(\mathbf{z}_C)$. $\qquad \square$

We list Zimmermann et al. (2021, Proposition 5.) for future use in our proof:

**Proposition 1** (Proposition 5 of Zimmermann et al. (2021).)**.** *Let $\mathcal{M}, \mathcal{N}$ be simply connected and oriented $\mathcal{C}^1$ manifolds without boundaries and $h : \mathcal{M} \to \mathcal{N}$ be a differentiable map. Further, let the random variable $\mathbf{z} \in \mathcal{M}$ be distributed according to $\mathbf{z} \sim p(\mathbf{z})$ for a regular density function $p$, i.e., $0 < p < \infty$. If the push-forward $p_{\#h}(\mathbf{z})$ through $h$ is also a regular density, i.e., $0 < p_{\#h} < \infty$, then $h$ is a bijection.*

**Theorem 3.2** (Identifiability from a *Set* of Views)**.** *Consider a set of views $\mathbf{x}_V$ satisfying Asm. 2.1, and let $G$ be a set of content encoders (Defn. 3.1) that minimizes the following objective*

$$\mathcal{L}(G) = \underbrace{\sum_{k < k' \in V} \mathbb{E}\left[\|g_k(\mathbf{x}_k) - g_{k'}(\mathbf{x}_{k'})\|_2\right]}_{\text{Content alignment}} - \underbrace{\sum_{k \in V} H\left(g_k(\mathbf{x}_k)\right)}_{\text{Entropy regularization}}, \tag{3.1}$$

*where the expectation is taken w.r.t. $p(\mathbf{x}_V)$ and $H(\cdot)$ denotes differential entropy. Then the shared **content** variable $\mathbf{z}_C := \{\mathbf{z}_j : j \in C\}$ is block-identified (Defn. 2.3) by $g_k \in G$ for any $k \in V$.*

*Proof.* Lemma C.1 verifies the existence of such a set of smooth encoders that obtains the global minimum of eq. (3.1) zero; Lemma C.2 derives the invariance conditions and the uniformity that the learned representations $g_k(\mathbf{x}_k)$ have to satisfy for all views $k \in V$. Based on the invariance condition eq. (C.5), Lemma C.3 shows that the learned representation $g_k(\mathbf{x}_k)$, $k \in V$ can only depend on the content block, not on any style variables, namely $g_k(\mathbf{x}_k) = h_k(\mathbf{z}_C)$ for some smooth function $h_k : \mathcal{Z}_C \to (0, 1)^{|C|}$.

We now apply Zimmermann et al. (2021, Proposition 5.) to show that all of the functions $h_k, k \in V$ are bijections. Note that both $\mathcal{Z}_C$ and $(0, 1)^{|C|}$ are simply connected and oriented $\mathcal{C}^1$ manifolds, and $h_k$ are smooth, thus differentiable, functions that map the intersection set of random variables $\mathbf{z}_C$ from $\mathcal{C}$ to $(0, 1)^{|C|}$. Given by Asm. 2.1(ii) that $p_{\mathbf{z}_C}$ and the push-forward function through $h_k$ (uniform distributions) are regular densities, we conclude that all $h_k$ are diffeomorphisms for all $k \in V$.

Thus we have shown that any content set of encoders $G$ that minimizes $\mathcal{L}(G)$ (eq. (3.1)) can extract the ground-truth content variables $\mathbf{z}_C$ from view $\mathbf{x}_k \in \mathcal{X}_k$ up to a bijection $h_k : \mathcal{Z}_C \to (0, 1)^{|C|}$:

$$g_k(\mathbf{x}_k) = h_k(\mathbf{z}_C), \tag{C.21}$$

That is, shared content $\mathbf{z}_C$ is block-identified by the content encoders $G = \{g_k\}_{k \in V}$. □

**Remark on the proof technique for Thm. 3.2.** For Thm. 3.2, one could imagine an alternate proof by induction over the number of views, where the proofs by von Kügelgen et al. (2021); Daunhawer et al. (2023) would be the base case. We opted for a direct proof technique as the induction proof may have been perhaps more intuitive at a high level but was significantly longer. Additionally, we present the current version because it would be generally more accessible as a more familiar proof technique.

C.2 PROOF FOR THM. 3.8

Our proof consists of the following steps:

1. We show in Lemma C.4 the loss eq. (C.22) is lower bounded by zero and construct optimal $R^*$ (Defn. 3.3), $\Phi^*$ (Defn. 3.5), $T^*$ (Defn. 3.6) that reach this lower bound;

2. Next, we show in Lemma C.6 that, if the content sizes $|C_i|$ are known for all $V_i \in \mathcal{V}$, then any view-specific encoders, content selectors, and projections $(R, \Phi, T)$ that minimize the loss eq. (C.22), block-identify the content variables $\mathbf{z}_{C_i}$ for any $V_i \in \mathcal{V}$, using similar steps as in the proof for Thm. 3.2.

3. As the third step, we show that any minimizer $R$ (Defn. 3.3), $\Phi$ (Defn. 3.5), $T$ (Defn. 3.6) of eq. (C.22) also minimizes the information-sharing regularizer (Defn. 3.7); and show that the optimal solution $(R^*, \Phi^*, T^*)$ we constructed in the first step reaches this lower bound of Defn. 3.7.

4. Then, we show *by contradiction* that any optimal content selector $\Phi^*$ that solves the constrained optimization problem in eq. (3.2) recovers the correct content size $|C_i|$ for each subset $V_i$, using the invariance condition in Lemma C.5.

5. Lastly, we apply the results from Lemma C.6 and conclude our proof for Thm. 3.8.

We rephrase each step as a separate lemma and use them to complete the final proof for Thm. 3.8.

**Lemma C.4** (Existence of Encoders, Selectors and Projections). *Consider a jointly observed set of views $\mathbf{x}_V$ satisfying Asm. 2.1. For any set of view-specific encoders $R$ (Defn. 3.3), content selectors $R_\Phi$ (Defn. 3.5) and projections $T$ (Defn. 3.6), we define the following objective:*

$$\mathcal{L}(R, \Phi, T) = \sum_{V_i \in \mathcal{V}} \sum_{\substack{k,k' \in V_i \\ k < k'}} \mathbb{E}\left[\left\|\phi^{(i,k)} \oslash r_k(\mathbf{x}_k) - \phi^{(i,k')} \oslash r_{k'}(\mathbf{x}_{k'})\right\|_2\right] - \sum_{k \in V} H\left(t_k \circ r_k(\mathbf{x}_k)\right). \tag{C.22}$$

*which is lower bounded by zero; and there exists such combination of $R, \Phi, T$ that obtains this global minimum zero.*

*Proof.* Consider the objective function $\mathcal{L}(R, \Phi, T)$ (eq. (C.22)), the global minimum of $\mathcal{L}(R, \Phi, T)$ is obtained when the first term (alignment) is minimized and the second term (entropy) is maximized. The alignment term is minimized to zero when selected representations $\phi^{(i,k)} \oslash r_k$ are perfectly aligned for all $k \in V$ almost surely. The second term (entropy) is maximized to zero *only* when $t_k \circ r_k(\mathbf{x}_k)$ is uniformly distributed on $(0,1)^{|S_k|}$ for all view $k \in V$. Thus we have shown that the loss (eq. (C.22)) is lower-bounded by zero.

The optimal view-specific encoders can be defined via the inverse of the view-specific mixing functions $\{f_k\}_{k \in V}$, which by Asm. 2.1(i) are smooth and invertible. By definition, we have $f_k^{-1}(\mathbf{x}_k) = \mathbf{z}_{S_k}$ for all $k \in V$. Formally, we define the set of optimal view-specific encoders

$$R := \{f_k^{-1}\}_{k \in V}. \tag{C.23}$$

Next, we define the optimal auxiliary transformation $t_k$ for each view $k$ using *Darmois construction*, writing $t_k \circ r_k(\mathbf{x}_k) = t_k \circ f_k^{-1}(\mathbf{x}_k) = t_k^j(\mathbf{z}_{S_k})$, we have:

$$t_k^j(\mathbf{z}_{S_k}) := F_j^k\left([\mathbf{z}_{S_k}]_j | [\mathbf{z}_{S_k}]_{1:j-1}\right) = \mathbb{P}\left([Z_{S_k}]_j \le [\mathbf{z}_{S_k}]_j | [\mathbf{z}_{S_k}]_{1:j-1}\right) \quad j \in \{1, \ldots, |S_k|\}, \tag{C.24}$$

where $F_j^k$ denotes the conditional cumulative distribution function (CDF) of $[\mathbf{z}_{S_k}]_j$ given $[\mathbf{z}_{S_k}]_{1:j-1}$. Thus, $t_k(\mathbf{z}_{S_k})$ is uniformly distributed on $(0,1)^{|S_k|}$ and $t_k$ is smooth by Asm. 2.1(ii) which states that $p_\mathbf{z}$ is a smooth density.

As for the optimal content selectors $\Phi = \{\phi^{(i,k)}\}_{V_i \in \mathcal{V}, k \in V_i}$, choose $\phi^{(i,k)}$ such that

$$\phi^{(i,k)} \oslash \hat{\mathbf{z}}_{S_k} := \hat{\mathbf{z}}_{C_i} \tag{C.25}$$

Writing $f_k^{-1}(\mathbf{x}_k) = \mathbf{z}_{S_k}$, the loss $\mathcal{L}(R, \Phi, T)$ from eq. (C.22) takes the value:

$$\begin{aligned}
\mathcal{L}(R, \Phi, T) &= \sum_{V_i \in \mathcal{V}} \sum_{\substack{k,k' \in V_i \\ k < k'}} \mathbb{E}\left[\left\|\phi^{(i,k)} \oslash r_k(\mathbf{x}_k) - \phi^{(i,k')} \oslash r_{k'}(\mathbf{x}_{k'})\right\|_2\right] - \sum_{k \in V} H\left(t_k \circ r_k(\mathbf{x}_k)\right) \\
&= \sum_{V_i \in \mathcal{V}} \sum_{\substack{k,k' \in V_i \\ k < k'}} \mathbb{E}\left[\left\|\phi^{(i,k)} \oslash f_k^{-1}(\mathbf{x}_k) - \phi^{(i,k')} \oslash f_{k'}^{-1}(\mathbf{x}_{k'})\right\|_2\right] - \sum_{k \in V} H\left(t_k \circ f_k^{-1}(\mathbf{x}_k)\right) \\
&= \sum_{V_i \in \mathcal{V}} \sum_{\substack{k,k' \in V_i \\ k < k'}} \mathbb{E}\left[\left\|\phi^{(i,k)} \oslash \mathbf{z}_{S_k} - \phi^{(i,k')} \oslash \mathbf{z}_{S_{k'}}\right\|_2\right] - \sum_{k \in V} H\left(t_k(\mathbf{z}_{S_k})\right) \\
&= \sum_{V_i \in \mathcal{V}} \sum_{\substack{k,k' \in V_i \\ k < k'}} \mathbb{E}\left[\left\|\mathbf{z}_{C_i} - \mathbf{z}_{C_i}\right\|_2\right] - \sum_{k \in V} H\left(t_k(\mathbf{z}_{S_k})\right) \\
&= 0
\end{aligned} \tag{C.26}$$

Note that the first term is minimized to zero because the shared content values $\mathbf{z}_{C_i}$ align among the views in one subset $V_i \in \mathcal{V}$; the second term is maximized to zero because $t_k(\mathbf{z}_{S_k})$ is uniformly distributed on $(0,1)^{|S_k|}$ given by the property of *Darmois construction* (Darmois, 1951). To this end, we have shown that there exists such optimum $R, \Phi, T$ as defined in eqs. (C.23) to (C.25) that minimizes the objective in eq. (C.22). $\square$

**Lemma C.5** (Conditions of Optimal Encoders, Selectors and projections). *Given the same set of views $\mathbf{x}_V$ as introduced in Lemma C.4, to minimize $\mathcal{L}(R, \Phi, T)$ in eq. (C.22), any optimum $R, \Phi, T$ (Defns. 3.3, 3.5 and 3.6) has to satisfy similar **invariance** and **uniformity** conditions from Lemma C.2:*

- *Invariance: All **selected** representations $\phi^{(i,k)} \oslash r_k(\mathbf{x}_k), k \in V$ must align across the views from the set $V_i \in \mathcal{V}$ almost surely:*

$$\phi^{(i,k)} \oslash r_k(\mathbf{x}_k) = \phi^{(i,k')} \oslash r_{k'}(\mathbf{x}_{k'}) \quad \forall V_i \in \mathcal{V} \, \forall k, k' \in V_i \quad a.s. \tag{C.27}$$

- *Uniformity: All extracted representations $t_k \circ r_k(\mathbf{x}_k), k \in V$ must be uniformly distributed over the hyper unit-cube $(0, 1)^{|S_k|}$.*

*Proof.* The minimum of $\mathcal{L}(R, \Phi, T) = 0$ can only be obtained when both terms are zero. For the first term (alignment) to be zero, it is necessary that $\phi^{(i,k)} \oslash r_k(\mathbf{x}_k) = \phi^{(i,k')} \oslash r_{k'}(\mathbf{x}_{k'})$ almost surely for all $V_i \in \mathcal{V}$, $k, k' \in V_i$ w.r.t. the true generating process. The second term (entropy) is upper-bounded by zero; this maximum can only be obtained when the auxiliary encoding $t_k \circ r_k(\mathbf{x}_k), k \in V$ follows *uniformity*, as also indicated by Lemma C.2. $\square$

**Lemma C.6** (View-Specific Encoder for Identifiability Given Content Sizes). *Consider a jointly observed set of views $\mathbf{x}_V$ satisfying Asm. 2.1 and assume that the dimensionality of the **subset-specific** content $|C_i|$ is given for all subset $V_i \in \mathcal{V}$. We consider a special type of content selectors $\Phi$ with $\left\|\phi^{(i,k)}\right\|_0 = |C_i|$ for all $k \in V_i$. Let $R, T$ respectively denote some view-specific encoders (Defn. 3.3), and projections (Defn. 3.6), which jointly minimize the following objective together with the special content selectors $\Phi$:*

$$\mathcal{L}(R, \Phi, T) = \sum_{V_i \in \mathcal{V}} \sum_{\substack{k, k' \in V_i \\ k < k'}} \mathbb{E}\left[\left\|\phi^{(i,k)} \oslash r_k(\mathbf{x}_k) - \phi^{(i,k')} \oslash r_{k'}(\mathbf{x}_{k'})\right\|_2\right] - \sum_{k \in V} H\left(t_k \circ r_k(\mathbf{x}_k)\right). \tag{C.28}$$

*Then for any view $k \in V$, any subset of views $V_i \in \mathcal{V}$ with $k \in V_i$, the composed function $\phi^{(i,k)} \oslash r_k$ block-identifies the shared **content** variables $\mathbf{z}_{C_i}$ in the sense that the learned representation $\hat{\mathbf{z}}_k^{(i)} := \phi^{(i,k)} \oslash r_k(\mathbf{x}_k)$ is related to the ground truth content variables through some smooth invertible mapping $h_k : \mathcal{Z}_{C_i} \to \mathcal{Z}_{C_i}$ with $\hat{\mathbf{z}}_k^{(i)} = h_k^{(i)}(\mathbf{z}_{C_i})$.*

*Proof.* Lemma C.4 verifies that there exists such optimum which minimizes the loss eq. (C.28) to zero; the invariance and uniformity conditions have to be satisfied by any optimum, as shown in Lemma C.5. Following Lemma C.3, the composition $r_k^{(i)} := \phi^{(i,k)} \oslash r_k$ can only encode information related to the subset-specific content $C_i$ for any subset $V_i \in \mathcal{V}$ otherwise it will lead to a contradiction to the invariance condition from Lemma C.5. The last step is to prove the invertibility of the encoders $G$. Notice that

$$t_k \circ r_k(\mathbf{x}_k) = t_k \circ r_k \circ f_k(\mathbf{z}_{S_k})$$

By applying Zimmermann et al. (2021, Proposition 5.) with similar arguments as in the proof for Thm. 3.2, we can show that composition $t_k \circ r_k \circ f_k$ is a smooth bijection of the subset-specific content $\mathbf{z}_{C_i}$. Since $f_k$ is a smooth invertible mapping by Asm. 2.1 (i), we have:

$$(t_k \circ r_k \circ f_k) \circ f_k^{-1} = (t_k \circ r_k) \circ (f_k \circ f_k^{-1}) = t_k \circ r_k,$$

Hence, $t_k \circ r_k$ is bijective as the composition of bijections is a bijection. Next, we show that $r_k$ is bijective. Showing that $r_k$ is bijective on its image is equivalent to showing that it is injective. By contradiction, suppose $r_k$ is not injective. Thus there exists distinct values $\mathbf{x}_k^1, \mathbf{x}_k^2 \in \mathcal{X}_k$ s.t. $r_k(\mathbf{x}_k^1) = r_k(\mathbf{x}_k^2)$. This implies that $t_k \circ r_k(\mathbf{x}_k^1) = t_k \circ r_k(\mathbf{x}_k^2)$, which violate injectivity of $t_k \circ r_k$. Thus, $r_k$ must be injective.

To this end, we conclude that any $R, \Phi, T$ that minimizes eq. (C.28) block-identifies the shared content variables $\mathbf{z}_{C_i}$ for any subset of views $V_i \in \mathcal{V}$. $\square$

**Claim 1.** For any $(R, \Phi, T)$ $(Defns. 3.1, 3.3 \text{ and } 3.6)$ that minimizes the loss eq. (C.22), the $\text{Reg}(\Phi)$ (Defn. 3.7) is lower bounded by $-\sum_{V_i \in \mathcal{V}} |C_i| \cdot |V_i|$ and this minimum is obtained at the optimal content selectors defined in eq. (C.25).

*Proof.* Suppose *for a contradiction* that there exists some binary weight parameters $\tilde{\Phi} \neq \Phi$ with

$$\text{Reg}(\tilde{\Phi}) = -\sum_{V_i \in \mathcal{V}} \sum_{k \in V_i} \left\| \tilde{\phi}^{(i,k)} \right\|_0 < \text{Reg}(\Phi), \tag{C.29}$$

which means, there exists at least one vector $\tilde{\phi}^{(i,k)}$ for some view $k \in V$, subset $V_i \in \mathcal{V}$, such that

$$\tilde{\phi}^{(i,k)} \oslash r_k(\mathbf{x}_k) = \hat{\mathbf{z}}_A \qquad |A| > |C_i|, \tag{C.30}$$

where $A \subseteq S_k$ is an index subset of the view-specific latents $S_k$. Given that $R, \Phi, T$ minimizes $\mathcal{L}(R, \Phi, T)$ from eq. (C.22), these minimizers have to satisfy the invariance and uniformity constraint as shown in Lemma C.5. Since uniformity implies invertibility (Zimmermann et al., 2021), the learned representation $r_k(\mathbf{x}_k)$ contains sufficient information about the original view $\mathbf{x}_k$ s.t. the view $\mathbf{x}_k$ can be reconstructed by some decoder given enough capacity. Given that the number of selected dimensions $|A| > |C_i|$, at least one latent component $j \in A$ will contain information that is not jointly shared by $V_i$. That means the composition $r_k^{(i)} := \phi^{(i,k)} \oslash r_k$ encodes some information other than just content $C_i$. As shown in Lemma C.3, any dependency from the learned representation on non-content variables leads to contradiction to the invariance condition as derived in Lemma C.5. Therefore, the optimal content selectors $\Phi$ following the definition in eq. (C.25) must obtain the global minimum of the information-sharing regularizer (Defn. 3.7), which equals $-\sum_{V_i \in \mathcal{V}} \sum_{k \in V_i} |C_i|$. □

**Theorem 3.8** (View-Specific Encoder for Identifiability)**.** *Let $R, \Phi, T$ respectively be any view-specific encoders (Defn. 3.3), content selectors (Defn. 3.1) and projections (Defn. 3.6) that solve the following constrained optimization problem:*

$$\min \text{Reg}(\Phi) \qquad \textit{subject to:} \qquad R, \Phi, T \in \arg\min \mathcal{L}(R, \Phi, T) \tag{3.2}$$

*where*

$$\mathcal{L}(R, \Phi, T) = \sum_{V_i \in \mathcal{V}} \sum_{\substack{k,k' \in V_i \\ k < k'}} \underbrace{\mathbb{E}\left[\left\| \phi^{(i,k)} \oslash r_k(\mathbf{x}_k) - \phi^{(i,k')} \oslash r_{k'}(\mathbf{x}_{k'}) \right\|_2\right]}_{\textit{Content alignment}} - \sum_{k \in V} \underbrace{H(t_k \circ r_k(\mathbf{x}_k))}_{\textit{Entropy}},$$
$$\tag{3.3}$$

*Then for any subset of views $V_i \in \mathcal{V}$ and any view $k \in V_i$, $\phi^{(i,k)} \oslash r_k$ block-identifies (Defn. 2.3) the shared **content** variables $\mathbf{z}_{C_i}$, as defined in eq. (2.2).*

*Proof.* Lemma C.4 confirms that there exist view-specific encoders $R$, content selectors $\Phi$, and projections $T$ that obtain the minimum of the unregularized loss eq. (C.22) (equals zero); Additionally, any optimal $R, \Phi, T$ fulfills the invariance condition and uniformity (Lemma C.5) s.t. they obtain the global minimum zero. Using the invariance condition, Claim 1 substantiates that the optimal content selectors as defined in eq. (C.25) minimizes the regularization term (Defn. 3.7). We have thus shown that with $R, \Phi, T$ (as defined in eqs. (C.23) to (C.25)), eq. (3.2) obtains the global minimum.

Next, we show that the number of selected dimensions from each selector $\phi^{(i,k)}$, i.e., the $L_0$ norm of $\phi^{(i,k)}$, align with the size of the shared content $|C_i|$.

Among the content selectors that minimize the unregularized loss (eq. (C.22)), we consider some content selectors $\Phi^* \in \arg\min \text{Reg}(\Phi)$ that also minimize the information-sharing regulariser defined in Defn. 3.7, that is:

$$\text{Reg}(\Phi^*) = -\sum_{V_i \in \mathcal{V}} \sum_{k \in V_i} |C_i|.$$

Suppose *for a contradiction* that there exists a pair of binary selectors $(\phi^{(i,k)}, \phi^{(i',k')})$ with $\phi^{(i,k)} \in \{0,1\}^{|S_k|}$ and $\phi^{(i',k')} \in \{0,1\}^{|S_{k'}|}$ such that

$$\left\| \phi^{(i,k)} \right\|_0 > |C_i|; \qquad \left\| \phi^{(i,k')} \right\|_0 < |C_{i'}|, \tag{C.31}$$

which indicates that there exists at least one latent component $j \in S_k \setminus C_i$ being selected by $\phi^{(i,k)}$; similarly, this contradicts the invariance condition as shown in Lemma C.3. Hence, the number of dimensions selected by each $\phi^{(i,k)}$ has to equal the content size $|C_i|$.

At this stage, the problem setup is reduced to the case in Lemma C.6 where the size of the content variables $|C_i|$ are given for all subset of views $V_i \in \mathcal{V}$. Hence, applying Lemma C.6, we conclude that any $R, \Phi, T$ (Defns. 3.3, 3.5 and 3.6) that minimize eq. (3.2) block-identify the shared content variables $\mathbf{z}_{C_i}$ for any subset of views $V_i \in \mathcal{V}$ and for all views $k \in V_i$. □

### C.3 Proofs for Identifiability Algebra

Let $\mathbf{z}_{C_1}, \mathbf{z}_{C_2}$ be two sets of content variables indexed by $C_1, C_2 \subseteq [N]$ that are block-identified by some smooth encoders $g_1 : \mathcal{X}_1 \to \mathcal{Z}_{C_1}, g_2 : \mathcal{X}_2 \to \mathcal{Z}_{C_2}$, then it holds for $C_1, C_2$ that:

**Corollary 3.9** (Identifiability Algebra: Intersection). *The intersection $\mathbf{z}_{C_1 \cap C_2}$ can be block-identified.*

*Proof.* By the definition of block-identifiability, we construct two synthetic views using the learned representation from $\mathbf{x}_1$ and $\mathbf{x}_2$:

$$\begin{aligned} \mathbf{x}^{(1)} &:= g_1(\mathbf{x}_1) = h_1(\mathbf{z}_{C_1}) \\ \mathbf{x}^{(2)} &:= g_2(\mathbf{x}_2) = h_2(\mathbf{z}_{C_2}) \end{aligned} \tag{C.32}$$

for some smooth invertible mapping $h_k : \mathcal{Z}_{C_k} \to \mathcal{Z}_{C_k} \, k \in \{1, 2\}$. Applying the Thm. 3.2 with two views, we can block-identify the intersection $C_1 \cap C_2$ using this pair of views $(\mathbf{x}^{(1)}, \mathbf{x}^{(2)})$. □

**Corollary 3.10** (Identifiability Algebra: Complement). *If $C_1 \cap C_2$ is independent of $C_1 \backslash C_2$, then the complement $\mathbf{z}_{C_1 \backslash C_2}$ can be block-identified.*

*Proof.* Construct the same synthetic views $\mathbf{x}^{(1)}, \mathbf{x}^{(2)}$ as in the proof for Cor. 3.9. We then can consider the intersection $C_1 \cap C_2$ as the content variable and $C_1 \backslash C_2$ as the style variable from these two synthetic views $(\mathbf{x}^{(1)}, \mathbf{x}^{(2)})$. *Private Component Extraction* from Lyu et al. (Theorem 2. 2021) has shown that if the style variable is independent of the content, then the style variables can also be extracted up to a smooth invertible mapping. Therefore, we conclude that the complement $\mathbf{z}_{C_1 \backslash C_2}$ can also be block-identified. □

**Corollary 3.11** (Identifiability Algebra: Union). *If $C_1 \cap C_2$, $C_1 \backslash C_2$ and $C_2 \backslash C_1$ are mutually independent, then the union $\mathbf{z}_{C_1 \cup C_2}$ can be block-identified.*

*Proof.* We rephrase $C_1 \cup C_2$ as a union of the following disjoint parts:

$$C_1 \cup C_2 = (C_1 \cap C_2) \cup (C_1 \backslash C_2) \cup (C_2 \backslash C_1) \tag{C.33}$$

Following the definition from Cors. 3.9 and 3.10 have shown that:

$$\begin{aligned} \hat{\mathbf{z}}_\cap &:= h_\cap(\mathbf{z}_{C_1 \cap C_2}) \\ \hat{\mathbf{z}}_{1 \backslash 2} &:= h_{1 \backslash 2}(\mathbf{z}_{C_1 \backslash C_2}) \\ \hat{\mathbf{z}}_{2 \backslash 1} &:= h_{2 \backslash 1}(\mathbf{z}_{C_2 \backslash C_1}), \end{aligned} \tag{C.34}$$

By concatenate the learned representations, we define $h_\cup : \mathcal{Z}_{C_1 \cup C_2} \to \mathcal{Z}_{C_1 \cup C_2}$ as

$$h_\cup(\mathbf{z}_C, \mathbf{z}_{1 \backslash 2}, \mathbf{z}_{2 \backslash 1}) := [\hat{\mathbf{z}}_C, \hat{\mathbf{z}}_{1 \backslash 2}, \hat{\mathbf{z}}_{2 \backslash 1}] = h_\cup(\mathbf{z}_{C_1 \cup C_2}), \tag{C.35}$$

hence, the union $C_1 \cap C_2$ can be block-identified. □

## D Experimental Results

This section provides further details about the datasets and implementation details in § 5. The implementation is built upon the code open-sourced by Zimmermann et al. (2021); von Kügelgen et al. (2021); Daunhawer et al. (2023).

### D.1 Numerical Experiment – Theory Validation

**Data Generation.** For completeness, we summarize the setting of our numerical experiments. We generate synthetic data following Example 2.1, which we also report below. The latent variables are sampled from a Gaussian distribution $\mathbf{z} \sim \mathcal{N}(0, \Sigma_\mathbf{z})$, where possible *causal* dependencies are encoded through $\Sigma_\mathbf{z}$. Note that in this setting the ground truth causal variables will be related linearly to each other.

$$\begin{aligned} \mathbf{x}_1 &= f_1(\mathbf{z}_1, \mathbf{z}_2, \mathbf{z}_3, \mathbf{z}_4, \mathbf{z}_5), & \mathbf{x}_2 &= f_2(\mathbf{z}_1, \mathbf{z}_2, \mathbf{z}_3, \mathbf{z}_5, \mathbf{z}_6), \\ \mathbf{x}_3 &= f_3(\mathbf{z}_1, \mathbf{z}_2, \mathbf{z}_3, \mathbf{z}_4, \mathbf{z}_6), & \mathbf{x}_4 &= f_4(\mathbf{z}_1, \mathbf{z}_2, \mathbf{z}_4, \mathbf{z}_5, \mathbf{z}_6). \end{aligned} \tag{D.1}$$

**Implementation Details.** We implement each view-specific mixing function $f_k$, for each view $k = 1, 2, 3, 4$, using a 3-layer *invertible, untrainable* MLP (Haykin, 1994) with LeakyReLU (Xu

Table 3: **Linear Evaluation**: Mean Correlation Coefficients across multiple views.

| | $(\mathbf{x}_1, \mathbf{x}_2)$ | $(\mathbf{x}_1, \mathbf{x}_3)$ | $(\mathbf{x}_1, \mathbf{x}_4)$ | $(\mathbf{x}_2, \mathbf{x}_3)$ | $(\mathbf{x}_2, \mathbf{x}_4)$ | $(\mathbf{x}_3, \mathbf{x}_4)$ | $(\mathbf{x}_1, \mathbf{x}_2, \mathbf{x}_3)$ | $(\mathbf{x}_1, \mathbf{x}_2, \mathbf{x}_4)$ | $(\mathbf{x}_1, \mathbf{x}_3, \mathbf{x}_4)$ | $(\mathbf{x}_2, \mathbf{x}_3, \mathbf{x}_4)$ | $(\mathbf{x}_1, \mathbf{x}_2, \mathbf{x}_3, \mathbf{x}_4)$ |
|---|---|---|---|---|---|---|---|---|---|---|---|
| ind. | $0.887 \pm 0.000$ | $0.881 \pm 0.000$ | $0.882 \pm 0.000$ | $0.885 \pm 0.000$ | $0.886 \pm 0.000$ | $0.880 \pm 0.000$ | $0.853 \pm 0.000$ | $0.854 \pm 0.000$ | $0.846 \pm 0.000$ | $0.851 \pm 0.000$ | $0.786 \pm 0.000$ |
| dep. | $0.956 \pm 0.000$ | $0.880 \pm 0.002$ | $0.891 \pm 0.002$ | $0.795 \pm 0.002$ | $0.805 \pm 0.002$ | $0.805 \pm 0.002$ | $0.945 \pm 0.001$ | $0.969 \pm 0.001$ | $0.858 \pm 0.003$ | $0.744 \pm 0.003$ | $0.944 \pm 0.001$ |

et al., 2015)($\alpha = 0.2$). The weight parameters in the mixing functions are *randomly initialized*. For the *learnable* view-specific encoders, we use a 7-layer MLP with LeakyReLU ($\alpha = 0.01$) for each view. The encoders are trained using the Adam optimizer (Kingma & Ba, 2014) with *lr=1e-4*. All implementation details are summarized in Tab. 4.

**Additional Experiments.** We experiment on *causally dependent* synthetic data, generated by $\mathbf{z} \sim \mathcal{N}(0, \Sigma_{\mathbf{z}})$ with $\Sigma_{\mathbf{z}} \sim \text{Wishart}(0, I)$. The results are shown in Fig. 4. The rows denote the ground truth latent factors, and the columns represent the learned representation from different subsets of views. Each cell reports the $R^2$ score between the respective ground truth factors and the learned representation. For example, the cell with col=$\{\mathbf{x}_1, \mathbf{x}_2\}$ and row=$\mathbf{z}_1$ shows the $R^2$ score when trying to predict $\mathbf{z}_1$ using the learned representation from subset $\{\mathbf{x}_1, \mathbf{x}_2\}$. Since dependent style variables become predictable, as discussed in App. D.1, we aim to verify that the learned representation contains *all and only* content variables. In other words, it *block-identifies* the ground truth content factors. For that, we consider all the views $\{\mathbf{x}_1, \ldots, \mathbf{x}_4\}$ and train a linear regression from the *ground truth content variables* $\mathbf{z}_1, \mathbf{z}_2$ to the individual style variables $\mathbf{z}_3, \mathbf{z}_4, \mathbf{z}_5$. We report the coefficient of determination $R^2$ in Tab. 6. We observe that the $R^2$ values obtained from the ground truth content are highly similar to the ones in the last column of the heatmap (Fig. 4). Based on this, we have showcased that the learned representation indeed *block-identifies* the content variables.

**Additional Evaluation Metric.** We report the Mean Correlation Coefficient(MCC) (Khemakhem et al., 2020) on the numerical experiments. MCC has been used in several recent works on identifiability of causal representation learning (Buchholz et al., 2023; von Kügelgen et al., 2023), it measures the *component-wise linear* correlation up to permutations. A high MCC (close to one) indicates a clear 1-to-1 linear correspondence between the learned representation and the ground truth latents. We remark our theoretical framework considers block-identifiability, which could imply any type of bijective relation to the ground truth content variables, including nonlinear transformations. Nevertheless, we observe high MCC score on both independent and dependent cases, showing that the learned representation having a high *linear* correlation to the latent components which indicates stronger identifiability results.

### D.2    SELF-SUPERVISED DISENTANGLEMENT

**Datasets.** In this experiment, we test on *MPI-3D complex* (Gondal et al., 2019) and *3DIdent* (Zimmermann et al., 2021). Both are high-dimensional image datasets generated from *mutually independent latent factors*: *MPI-3D complex* contains real-world complex shaped images with ten *discretized* latent factors while *3DIdent* renders a teapot conditioned on ten *continuous* latent factors.

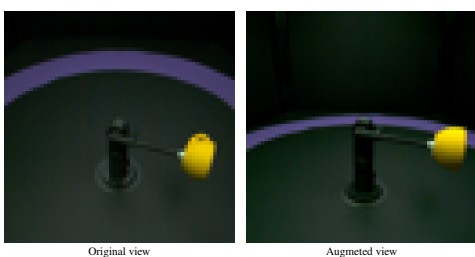
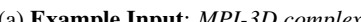
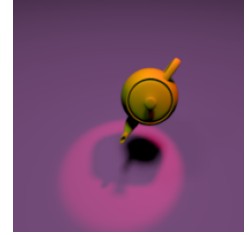
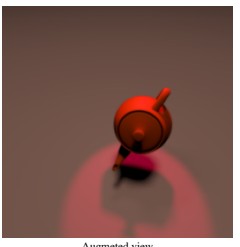

(a) **Example Input**: *MPI-3D complex*          (b) **Example Input**: *3DIdent*

**Implementation Details.** We used the implementation from (Michlo, 2021) for Ada-GVAE (Locatello et al., 2020), following the same architecture as (Locatello et al., 2020, Tab. 1 Appendix ). For our method, we use ResNet-18 (He et al., 2016) as the image encoder, details given in Tab. 8. For both approaches, we set ENCODING SIZE=10, following the setup in Locatello et al. (2020).

Table 4: Parameters for numerical simulation (§ 5.1 and App. D.1).

| Parameter | Value |
|---|---|
| Mixing function | 3-layer MLP |
| Encoder | 7-layer MLP |
| Optimizer | Adam |
| Adam: learning rate | 1e-4 |
| Adam: beta1 | 0.9 |
| Adam: beta2 | 0.999 |
| Adam: epsilon | 1e-8 |
| Batch size | 4096 |
| Temperature $\tau$ | 1.0 |
| # Iterations | 100,000 |
| # Seeds | 3 |
| Similarity metric | Euclidian |

Table 5: Parameters for experiments §§ 5.2 and 5.3 and Apps. D.3 and D.4. *: for both image and text encoders. **: hyper-arapmeter for BarlowTwins (Zbontar et al., 2021).

| Parameter | Values |
|---|---|
| Content encoding size* | 8 |
| View-specific encoding size* | 11 |
| Image hidden size | 100 |
| Text embedding dim | 128 |
| Text vocab size | 111 |
| Text fbase | 25 |
| Batch size | 128 |
| Temperature | 1.0 |
| Off-diagonal constant $\lambda$** | 1.0 |
| Optimizer | Adam |
| Adam: beta1 | 0.9 |
| Adam: beta2 | 0.999 |
| Adam: epsilon | 1e-8 |
| Adam: learning rate | 1e-4 |
| # Iterations | 300,000 |
| # Seeds | 3 |
| Similarity metric | Cosine similarity |
| Gradient clipping | 2-norm; max value 2 |

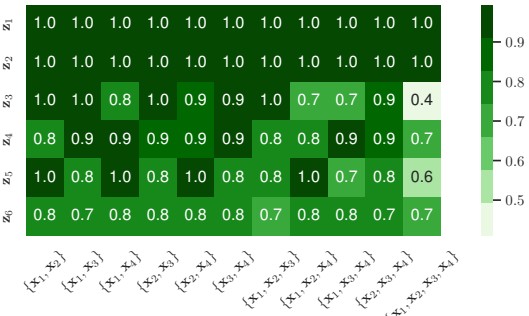

Figure 4: **Theory Verfication:** Average $R^2$ across multiple views generated from *causally dependent* latents.

Table 6: **Linear $R^2$ from *ground truth content variables to styles*** when consider $\{\mathbf{x}_1, \mathbf{x}_2, \mathbf{x}_3, \mathbf{x}_4\}$, these values align with the last column of Fig. 4, showing that we have *block-identified* the content variables $\{\mathbf{z}_1, \mathbf{z}_2\}$

| content | style | | | |
|---|---|---|---|---|
| $\{\mathbf{z}_1, \mathbf{z}_2\}$ | $\mathbf{z}_3$ | $\mathbf{z}_4$ | $\mathbf{z}_5$ | $\mathbf{z}_6$ |
| 1.0 | 0.32 | 0.65 | 0.58 | 0.71 |

Table 7: **Thm. 3.2 Validation** on *Causal3DIdent*: $R^2$ mean±std. *Green*: content, **bold**: $R^2 > 0.50$.

| Views generated by changing | class | positions | | | | hues | | | rotations | | |
|---|---|---|---|---|---|---|---|---|---|---|---|
| | | $x$ | $y$ | $z$ | spotl | obj | spotl | bkg | $\phi$ | $\theta$ | $\psi$ |
| hues | **1.00±0.00** | **0.76±0.01** | **0.56±0.02** | 0.00±0.00 | **0.82±0.01** | 0.27±0.03 | 0.00±0.01 | 0.00±0.00 | 0.25±0.02 | 0.27±0.02 | 0.27±0.02 |
| positions | **1.00±0.00** | 0.00±0.01 | 0.46±0.02 | 0.00±0.01 | 0.00±0.01 | 0.32±0.02 | 0.00±0.01 | **0.92±0.00** | 0.26±0.02 | 0.29±0.02 | 0.27±0.02 |
| rotations | **1.00±0.00** | 0.11±0.01 | **0.50±0.02** | 0.00±0.00 | 0.06±0.01 | 0.31±0.02 | 0.00±0.01 | **0.83±0.01** | 0.25±0.01 | 0.27±0.02 | 0.06±0.01 |
| hues+pos | **1.00±0.00** | 0.00±0.00 | 0.20±0.02 | 0.00±0.01 | 0.00±0.01 | 0.14±0.02 | 0.00±0.00 | 0.00±0.01 | 0.07±0.01 | 0.18±0.02 | 0.12±0.02 |
| hues+rot | **1.00±0.00** | 0.09±0.02 | 0.36±0.02 | 0.00±0.00 | **0.51±0.01** | 0.25±0.02 | 0.00±0.01 | 0.00±0.01 | 0.00±0.01 | 0.25±0.02 | 0.25±0.01 |
| pos+rot | **1.00±0.00** | 0.00±0.00 | 0.21±0.02 | 0.00±0.01 | 0.00±0.00 | 0.07±0.01 | 0.00±0.01 | 0.23±0.02 | 0.05±0.01 | 0.20±0.02 | 0.13±0.02 |
| hues+pos+rot | **1.00±0.00** | 0.00±0.00 | 0.42±0.02 | 0.00±0.01 | 0.00±0.00 | 0.25±0.02 | 0.00±0.00 | 0.00±0.00 | 0.01±0.01 | 0.26±0.02 | 0.26±0.02 |

### D.3 CONTENT-STYLE IDENTIFIABILITY ON IMAGES

**Datasets.** *Causal3DIdent* (von Kügelgen et al., 2021) extends *3Dident* (Zimmermann et al., 2021) by introducing different classes of objects, thus *object shape* (or *class*) is added as an additional *discrete* factor of variation. We extend the image pairs experiments from (von Kügelgen et al., 2021) by inputting three views, as shown in Fig. 6b, where the second and third images are obtained by perturbing different subsets of latent factors of the first image. To perturb one specific latent component, we *uniformly* sample one latent in the predefined latent space ($Unif[-1, 1]$, details see (von Kügelgen et al., 2021, App. B)), then we use indexing search to retrieve the image in the dataset that has the closest latent values as the sampled ones. Note that only a finite number of images are available; thus, there is not always a perfect match. More frequently, we observe slight changes in the non-perturbing latent dimensions. For instance, the *hues* of the third view is slightly different than the original view, although we intended to share the same *hue* values.

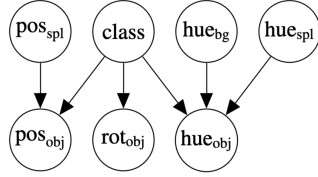

(a) **Underlying causal relation** in *Causal3DIdent* and *Multimodal3DIdent* images. Figure adopted from (von Kügelgen et al., 2021, Fig. 2)

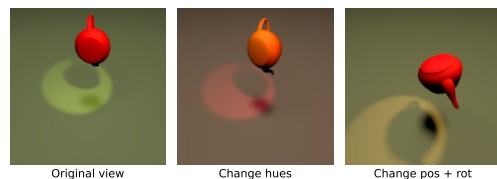

(b) **Example Input**: *Causal3DIdent*

Figure 6: *Causal3DIdent*: Underlying causal relations and input examples.

**Implementation Details.** The encoder structure and parameters are summarized in (Tabs. 5 and 8). We train using BarlowTwins (Zbontar et al., 2021) with cosine similarity and off-diagonal importance constant $\lambda = 1$. BarlowTwins is another contrastive loss that jointly encourages alignment and uniformity, by enforcing the cross-correlation matrix of the learned representations to be identity. The on-diagonal elements represent the *content alignment* while the off-diagonal elements approximate the *entropy regularization*.

**Thm. 3.2 Validation.** We train *content encoders* (Defn. 3.1) on *Causal3DIdent* to verify Thm. 3.2. Note that we experiment on three views (von Kügelgen et al., 2021) cannot naively handle. Tab. 7 summarizes results for all possible perturbations among the three views. We can observe that the discrete factor *class* learned perfectly; dependent style variables become predictable from the content (*class*) latent causal dependence. Note that this table shows similar results as in (von Kügelgen et al., 2021, Table 6. Latent Transformation (LT)). We remark that there is a reality between theory in practice: In theory, we assume that the content variables share the *exact same value* across all views; however, in practice, finding a perfect match of all of the *continuous* content values become impossible, since there is only a finite number of training data available. We believe this reality gap negatively influenced the learning performance on the content variables, thus preventing efficient prediction on certain content variables, such as *object hues*.

### D.4 MULTI-MODAL CONTENT-STYLE IDENTIFIABILITY UNDER PARTIAL OBSERVABILITY

**Dataset.** *Multimodal3DIdent* (Daunhawer et al., 2023) augments *Causal3DIdent* (von Kügelgen et al., 2021) with text annotations for each image view, and discretizes the *objection positions* $(x, y, z)$

Table 8: **Encoder Architectures** for *Causal3DIdent* and *Multimodal3DIdent*.

| Image Encoder | Text Encoder |
|---|---|
| *Input size = H × W × 3* | *Input size = vocab size* |
| ResNet-18(hidden size) | Linear(fbase, text embedding dim) |
| LeakyReLu($\alpha = 0.01$) | Conv2D(1, fbase, 4, 2, 1, bias=True) |
| Linear(hidden size, image encoding size) | BatchNorm(fbase); ReLU |
| | Conv2D(fbase, fbase·2, 4, 2, 1, bias=True) |
| | BatchNorm(fbase·2); ReLU |
| | Conv2D(fbase·2, fbase·4, 4, 2, 1, bias=True) |
| | BatchNorm(fbase·4); ReLU |
| | Linear(fbase·4·3·16, text encoding size) |

Table 9: **Thm. 3.2 Validation on** *Multimodal3DIdent*: $R^2$ mean±std. *Green*: content, **bold**: $R^2 > 0.50$.

| views generated by changing | class | img pos | | | img hues | | | txt class | txt pos | | txt hue | txt phrasing |
|---|---|---|---|---|---|---|---|---|---|---|---|---|
| | | $x$ | $y$ | spotl | obj | spotl | bkg | | $x$ | $y$ | obj_color_idx | |
| hues + rot | **0.82 ±0.01** | **1.00 ±0.00** | **1.00 ±0.00** | 0.00 ±0.00 | **0.87 ±0.01** | 0.00 ±0.00 | - | **0.85 ±0.03** | **1.00 ±0.00** | **1.00 ±0.00** | 0.15 ±0.02 | 0.21 ±0.02 |
| pos | **1.00 ±0.00** | 0.47 ±0.02 | **0.64 ±0.01** | 0.00±0.00 | **0.67 ±0.02** | 0.00 ±0.00 | - | **1.00 ±0.00** | 0.34 ±0.02 | **0.94±0.01** | 0.16 ±0.03 | 0.21 ±0.02 |

to categorical variables. In particular, *object-zpos* is a constant and thus not shown in our evaluation (Fig. 3). Our experiment extends (Daunhawer et al., 2023) by adding one additional image to the original image-text pair, perturbing the *hues*, *object rotations* and *spotlight positions* of the original image (Uniformly sample from $Unif[0, 1]$). Thus, $(\text{img}_0, \text{img}_1)$ share *object shape* and *background color*; Thus, $(\text{img}_0, \text{txt}_0)$ share *object shape* and *object x-y positions*; both $(\text{img}_1, \text{txt}_0)$ and the joint set $(\text{img}_0, \text{img}_1, \text{txt}_0)$ share only the *object shape*. One example input is shown in Fig. 7.

**Implementation Details.** Tabs. 5 and 8 shows the network architecture and implementation details, mostly following (Daunhawer et al., 2023). Note that we use the same encoding size for both image and text encoders for the convenience of implementation. We train using *BarlowTwins* with $\lambda = 1$. In practice, we treat the *unknown content sizes* as a list of hyper-parameters and optimize it over different validations.

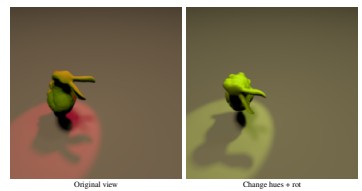

*Text description:* A **hare** of **bright yellow green** color is visible, positioned at the **mid-left** of the image.

Figure 7: **Example input**: *Multimodal3DIdent*. *Left:* pair of images that are original and perturbed images. *Right:* Text annotation for the original view.

**Further Discussion about § 5.3.** The fundamental difference between the *Multimodal3D* and *Causal3DIdent* datasets, as mentioned above, makes a direct comparison between our results in Fig. 3 and (von Kügelgen et al., 2021) harder. However, Tab. 7 shows similar $R^2$ scores as the results given in von Kügelgen et al. (2021, Sec 5.2), which verifies the correctness of our method.

**Thm. 3.2 Validation.** We additionally learn *content encoders* on three partially observed views $(\text{img}_0, \text{img}_1, \text{txt}_0)$ using the loss from Zbontar et al. (2021), to justify Thm. 3.2. We use the same architecture and parameters as summarized in Tabs. 5 and 8. Tab. 9 shows the content encoders consistently predict the content variables well and that our evaluation results highly align with (Daunhawer et al., 2023, Fig. 3) on image-text pairs $(\text{img}_0, \text{txt}_0)$ as inputs, which empirically validates Thm. 3.2.

### D.5 MULTI-TASK DISENTANGLEMENT WITH SPARSE CLASSIFIERS

Following Example 2.1, we synthetically generate the class labels by linear/nonlinear labeling functions on the shared content values, which resembles the underlying inductive bias from (Lachapelle et al., 2023; Fumero et al., 2023), that the shared features across different images with the same label should be most task-relevant. Here, we use the same sparse-classifier implementation from (Fumero et al., 2023). We remark that the goal of this experiment is to verify our expectation from Thm. 3.2

that the method of (Fumero et al., 2023) can be explained by our theory, although they assume mutually independent latents, which is a special case of our setup. In our experimental setup, an input gets a label 1 when the following labeling function value is greater than zero:

- Linear: $\sum_{j=1}^{d} \hat{\mathbf{z}}_j$ where $d$ denotes the encoding size.

- Nonlinear: $\tanh\left(\sum_{j=1}^{d} \hat{\mathbf{z}}_j^3\right)$

Thus, we have resembled the inductive hypothesis in Fumero et al. (2023) that the classification task is *only* dependent on the shared features. The fact that the binary classification is solved in several iterations verifies that Fumero et al. (2023) used the same *soft* alignment principle as described in Thm. 3.2.

## E  DISCUSSION

**The Theory-Practice Gap**   It is noticeable that some of the technical assumptions we made in the theory may not exactly hold in practice. A common assumption in identifiability literature is that the latent variables $\mathbf{z}$ are continuous, while this is not true for e.g. the *object shape* in *Causal3DIdent* (von Kügelgen et al., 2021) and *object shape, positions* in *Multimodal3DIdent* (Daunhawer et al., 2023). Another related gap regarding the dataset is that the additional views are generated by uniformly sampling a subset of latents from the original view and then trying to retrieve an image among the *existing* finite dataset, whose latent value is closest to the proposed one. However, having only a finite number of images implies that always finding a perfect match for each perturbed latents is almost impossible in practice. As a consequence, the designed to be strictly aligned content values between different views could differ from each other by a certain margin. Also, both Thms. 3.2 and 3.8 holds asymptotically and the global minimum is obtained only when given infinitely amount of data. Given that there is no closed-form solution for *entropy regularization* term eqs. (3.1) and (3.2), it is approximated either using negative samples (Oord et al., 2018; Chen et al., 2020) or by optimizing the cross-correlation between the encoded information to be close to Identity matrix (Zbontar et al., 2021); in both cases there is only a finite number of samples available, which makes converging to global minimum almost impossible in practice.

**Discovering Hidden Causal Structure from Overlapping Marginals.** Identifying latent blocks $\{\mathbf{z}_{B_i}\}$ provides us with access to the marginal distributions over the corresponding subsets of latents $\{p(\mathbf{z}_{B_i})\}$. With observed variables and known graph, this has been termed the "causal marginal problem" (Gresele et al., 2022), and our setup could therefore also been seen as generalization along those dimensions. It may be possible to extract some causal relations from the inferred marginal distributions over blocks, either by imposing additional assumptions or through constraint-based approaches (Triantafillou et al., 2010).

**How to Learn the Content Sizes?**   Thm. 3.8 shows that content blocks from any arbitrary subset of views can be discovered simultaneously using view-specific encoders (Defn. 3.3), content selectors (Defn. 3.5) and some projections (Defn. 3.6). We remarked in the main text that optimizing the information-sharing regularizer (Defn. 3.7) is highly non-convex and thus impractical. We proposed alternatives for both unsupervised and supervised cases: for self-supervised representation learning, one could employ *Gumble-Softmax* to learn the hard mask. We hypothesize that if there is an additional inclusion relation about the content blocks available, for example, we know that $C_1 \subseteq C_2 \subseteq C_3$, then the learning process could be eased by coding this inclusion relation in the mask implementation. This additional information is naturally inherited from the fact that the more views we include, the smaller the shared content will be. Another idea would be manually allocating individual content blocks in the learned encoding in a sequential manner, e.g. we set index=1, 2, 3 for the first content block and index=4, 5 for the second content block, and enforcing alignment correspondingly. Thus, for each view, we learn a concatenated representation of the shared content. Although this method does not perfectly follow the theoretical setting in the Thm. 3.8, it still learn all of the contents simultaneously and shows faster convergence. In classification tasks, the hard alignment constraint is relaxed to some soft constraint within one equivalence class e.g. samples which have the same label. In this case, we can replace the binary content selector with linear readouts, as studied and implemented by Fumero et al. (2023). Another, yet the most common approach to deal with this problem is to treat the content sizes as hyperparameters, as shown in (von Kügelgen et al., 2021; Daunhawer et al., 2023).

**Trade-off between Invertibility and Feature Sharing.** An invertible encoder implies that the extracted representation is a lossless compression, which means that the original observation can be reconstructed using this learned representation, given enough capacity. On the one hand, the invertibility of the encoders is enforced by the *entropy regularization*, such that the encoder preserves all information about the content variables; on the other hand, the info-sharing regularizer (Defn. 3.7) encourages reuse of the learned feature, which potentially prevents perfect reconstruction for each individual view. Intuitively, Thm. 3.8 seeks the sweet spot between invertibility and feature sharing: When the encoder shares more than the ground truth, then it loses information about certain views, and thus the compression is not lossless; When the invertibility is fulfilled but the info-sharing is not maximized, then the learned encoder is not an optimal solution either, given by the regularization penalty from the infor-sharing regularizer.

**Causal Representation Learning from Interventional Data.** Our framework considers purely *observational* data, where multiple partial views are generated from concurrently sampled latent variables using view-specific mixing functions. Recent works (Liang et al., 2023; Buchholz et al., 2023; von Kügelgen et al., 2023) have shown identifiability in non-parametric causal representation learning using interventional data, allowing discovering (some) hidden causal relations. Since simultaneously identifying the latent representation and the underlying causal structure in a *partially observable* setup has been a long standing goal, we believe incorporating interventional data into the proposed framework could be one interesting direction to explore.

