# OpenReview forum: "Multi-View Causal Representation Learning with Partial Observability"
_ICLR.cc/2024/Conference — ICLR 2024 spotlight_

### Official Review · Reviewer_BPer · 2023-10-31

**Soundness:** 3 good
**Presentation:** 2 fair
**Contribution:** 2 fair
**Rating:** 6
**Confidence:** 3

**Summary:**

This paper generalizes existing theories on causal representation learning, multi-view learning by allowing multi-views, partial observability and dependency between latents. It provides identifiability theory for content variables and so-called identifiability algebra to combine content variables.

**Strengths:**

1. The literature review is extensive, although there are some recent works on CRL including identification under the nonparametric settings that might be relevant as well.

[1] Liang, Wendong, et al. "Causal Component Analysis." arXiv preprint arXiv:2305.17225 (2023).
[2] Jiang, Yibo, and Bryon Aragam. "Learning nonparametric latent causal graphs with unknown interventions." arXiv preprint arXiv:2306.02899 (2023).
[3] Buchholz, Simon, et al. "Learning Linear Causal Representations from Interventions under General Nonlinear Mixing." arXiv preprint arXiv:2306.02235 (2023).

2. I like how the paper is structured with an intuition behind definitions and theorems for readability.

**Weaknesses:**

1. Although the setting is different, the theoretical contributions and proof techiniques seem to be mostly derived from [1] and [2].

[1] Roland S. Zimmermann, Yash Sharma, Steffen Schneider, Matthias Bethge, and Wieland Brendel. Contrastive learning inverts the data generating process. Proceedings of Machine Learning Research, 139:12979–12990, 2021

[2] Julius von Kügelgen, Yash Sharma, Luigi Gresele, Wieland Brendel, Bernhard Schölkopf, Michel Besserve, and Francesco Locatello. Self-supervised learning with data augmentations provably isolates content from style. Advances in neural information processing systems, 34:16451–16467, 2021

**Questions:**

1. For definition 5.3, why assume l0 to be the same and not the two vectors are the same? For a given view, the content variables are fixed, right?

2. Could you provide a little more motivation as to why view-specific encoders are needed? I understand the theoretical reason. But in practice, one would expect to train with all views available, right?

---

> ### Author Response · Authors · 2023-11-14
> **Response to Reviewer BPer**
>
> We thank the reviewer for the positive feedback about the structure of the paper. We also appreciate further references on identifiability under nonparametric settings [1, 2, 3], which we will discuss in the next revision. In the following we will answer the questions in detail.
>
> &nbsp;
> ### *Question 1:  why not assume two vectors are the same?*
>
>   - **The shared content can be indexed at different positions**, especially when the number of views grows.  Since we consider a partially observable setup, each observation is generated by a subset of latents and, in general, does not have access to all of the latents.
>
>   + **For example**, say there are three underlying latent variables $z_1, z_2, z_3$. and two observations $x_1=f_1(z_1, z_2)\, x_2 = f_2(z_2, z_3)$. In this case, the optimal selecting vectors for each should be $\phi_1 = (0, 1), \phi_2 = (1, 0)$, as different views are generated by different latents requiring different indexing.
>
> &nbsp;
> ### *Question 2: motivation of view-specific encoders*
>   - **Our work focuses on a multi-view and potentially multimodal setup**, where data are potentially collected by different measuring devices. Each view with a different modality would require a separate encoder. Even if two views are from the same modality, we can only share parameters if they measure the same latents.
>
>   + We remark that the **joint intersection** (content by considering all available views) **differs from the intersection of all subsets of views** (and could be empty). **For example**, we have three partial views $x_1 = f_1(z_1, z_2)$,  $x_2 = f_2(z_2, z_3)$, $x_3 = f_3(z_1, z_3)$. In this case, the **joint content is empty** while the pairwise intersection would give us the latent variables of interest: e.g. we can identify $z_2$ from $(x_1, x_2)$; $z_3$ from $(x_2, x_3)$ and $z_1$ from $(x_1, x_3)$.

---

> > ### Comment · Reviewer_BPer · 2023-11-21
> >
> > Thanks for your thoughtful rebuttal. I have decided to maintain my ratings.

---

> > > ### Author Response · Authors · 2023-11-23
> > > **Response to Reviewer BPer [2]**
> > >
> > > We sincerely thank the reviewer for keeping the recommendation of acceptance. We hope we can clarify some of your concerns, if further clarification is needed or if there are additional concerns, we are more than willing to engage in further discussion.

---

### Official Review · Reviewer_QCXK · 2023-10-31

**Soundness:** 3 good
**Presentation:** 4 excellent
**Contribution:** 2 fair
**Rating:** 6
**Confidence:** 3

**Summary:**

This work studies identifying latent causal representations given multi-view data. This work shows that the latent subspaces if shared by more than one view, can be identified up to bijjective mappings. Further, a view-specific encoder is devised to make learning a large number of shared blocks tractable. Experiments on multi-modality/task data are provided to verify the theoretical insights.

**Strengths:**

• The paper is well-written and communicates clearly motivations, formulation, technical details, and theoretical implications.

• This paper extends previous content-identification results (mainly von Kugelgen et al. 2021, Daunhawer et al. 2023)  to multi-view distributions and further (identifiability algebra).

• The empirical evaluation is thorough (over multi-view/modality/task datasets) and well-designed to substantiate the theoretical findings in the paper.

**Weaknesses:**

• My primary concern is the novelty of the work. In comparison with prior work (von Kugelgen et al. 2021; Daunhawer et al. 2023), the technical contribution appears somewhat incremental.

• The assumption that each view-specific generating function is invertible is strong, especially over multiple modalities. Intuitively, this requires the shared blocks to be duplicated multiple times and thus simplifies the identification problem. A recent work [1] has attempted to weaken this assumption.

[1] https://arxiv.org/abs/2306.07916

**Questions:**

I would like to learn about the authors' response to the weaknesses listed above, which may give me a clearer perspective on the paper's contribution.

---

> ### Author Response · Authors · 2023-11-14
> **Response to Reviewer QCXK**
>
> - We thank the reviewer for the valuable insights and the positive feedback on the written style and empirical evaluation. Our clarification regarding the novelty and contribution is listed in the *[general response](https://openreview.net/forum?id=OGtnhKQJms&noteId=qN34EnF5VV)*.
>
> + We also thank the reviewer for the reference, which we will discuss in the revised version. **The key difference** is that  [1] assumes structural conditions on the underlying causal graph (Condition 2.4) whereas our work **does not assume anything about the underlying causal structure**. Overall, we completely agree that adding structural assumptions while weakening the assumption on the mixing functions is an interesting direction for future exploration.

---

> > ### Comment · Reviewer_QCXK · 2023-11-21
> >
> > Thank you for taking the time to provide a thorough rebuttal and clarification. I appreciate the additional information and perspectives shared in your response. After careful consideration of your points and the initial review, I have decided to maintain the current rating.

---

> > > ### Author Response · Authors · 2023-11-23
> > > **Response to Reviewer QCXK [2]**
> > >
> > > We would be happy if we could address some of your concerns. Thank you very much for investing your time in the discussion, as well as your valuable feedback and the recommendation of acceptance.

---

### Official Review · Reviewer_rWVP · 2023-10-31

**Soundness:** 3 good
**Presentation:** 4 excellent
**Contribution:** 4 excellent
**Rating:** 8
**Confidence:** 4

**Summary:**

This work studies causal representation learning under partial observability. Causal representation learning is the emerging field of learning generative representations of data that could potentially have causal relationships among them. Identifiability refers to the existence of a unique generative model and is an important aspect of this field. Many works on CRL usually make certain simplifying functional assumptions or access to interventions, in order to exhibit identifiability. In this work, the authors provide a unified framework for identifiability in multi-view CRL, under various assumptions.

The data is assumed to be split into multiple views which are observed, where each view is a non-linear mixing function of a subset of the latent variables. Standard assumptions are made, including that the prior density is smooth and mixing functions are diffeomorphic. Under these assumptions, block-identifiability results are derived, when we have access to multiple partial views of the data. Because of the generality of the result, various prior results in this field can be viewed as special cases. The conceptual idea is that if we have a set of views, then we can identify a set of content encoders using a mix of an alignment and a regularization term. However, this leads to a potentially exponential number of encoders, which the authors handle via a single view-specific encoder. Experiments on synthetic data and visual/text data (Causal3DIdent and Multimodal 3DIdent) validate their theoretical ideas. The R^2 metric is reported, which measures how well the latents have been recovered. Overall, the paper is technically well-executed and fits the conference.

### References:

- [1] Learning Linear Causal Representations from Interventions under General Nonlinear Mixing

- [2] Nonparametric Identifiability of Causal Representations from Unknown Interventions

- [3] Identifiability of deep generative models without auxiliary information

- [4] Identifying Weight-Variant Latent Causal Models

**Strengths:**

- Causal representation learning has gotten a lot of attention recently, due to the promises it holds. This work is another important step in this direction.

- The general identifiability result captures a variety of prior works on CRL, therefore it serves as a neat unifying contribution.

- The approach to avoid using exponential set of encoders for each subset of views is very neat and is crucial for this work, for efficiency purposes.

**Weaknesses:**

- R^2 metric is reported for validating identifiability, however as the authors also note, there are instances when R^2 scores can be inflated. Why didn't the authors also cite other standard identifiability metrics like MCC (such as the ones reported in [1], [2])?

**Questions:**

Some questions were raised above. Additional suggestions:

- The recent work [1] also uses contrastive learning in CRL. Are there any resemblances to their approach, i.e. how do their loss function compare to content alignment?

- The works [3], [4] also study nonlinear ICA without auxiliary variables. I believe they use variants of VaDEs in their experiments, can this also be viewed as a soft alignment inductive bias?

---

> ### Author Response · Authors · 2023-11-14
> **Response to Reviewer rWVP**
>
> We appreciate the reviewer for providing further interesting references, which we will discuss in the revised version of the paper.
>
>   - **Relation with [1]**: Although [1] also focus on identifiability in causal representation learning, their setting is completely different in that they assume (unknown) **single-node** interventions which we believe is a very strict assumption. They generalize prior work that uses single-node interventions in the class of mixing functions. *Their cross-entropy loss is related to our alignment and entropy terms*. Incorporating the NOTEARS objective is something we did not explore and may be interesting to combine the causal representation learning with causal discovery. We will add a citation in the revised manuscript, highlighting this connection as interesting future works.
>
>   + **Relation with [3-4]**: We fully agree with the reviewer that there is a connection with non-linear ICA with auxiliary variables, which we discussed in section 4, paragraph 4. We will add these references in this section for the next revision.
>
> &nbsp;
>
>
> We also thank the reviewer for proposing another metric for identifiability. We remark that MCC measures **component-wise linear correlation** up to permutations, which differs from our setup: The definition of block-identifiability implies any type of bijective relation to the ground truth content variables, including **nonlinear** transformations, which MCC, in general, cannot capture. However, we fully agree with the reviewer that MCC is a very interesting metric that provides us more insight into the relation between the predicted and ground truth latents. Thus, we **report the MCC score** for the numerical experiments with independent (Sec 5.1) and dependent latent variables (Appendix D.1):
>
> |      | $(x_0, x_1)$    | $(x_0, x_2)$    | $(x_0, x_3)$    | $(x_1, x_2)$    | $(x_1, x_3)$    | $(x_2, x_3)$    | $(x_0, x_1, x_2)$ | $(x_0, x_1, x_3)$ | $(x_0, x_2, x_3)$ | $(x_1, x_2, x_3)$ | $(x_0, x_1, x_2, x_3)$ |
> | ----------- | ----------- | ----------- | ----------- | ----------- | ----------- | ----------- | ------------ | ------------ | ------------ | ------------ | ---------------- |
> | independent | 0.887±0.000 | 0.881±0.000 | 0.882±0.000 | 0.885±0.000 | 0.886±0.000 | 0.880±0.000 | 0.853±0.000  | 0.854±0.000  | 0.846±0.000  | 0.851±0.000  | 0.786±0.000      |
> | dependent   | 0.956±0.000 | 0.880±0.002 | 0.891±0.002 | 0.795±0.002 | 0.805±0.002 | 0.805±0.002 | 0.945±0.001  | 0.969±0.001  | 0.858±0.003  | 0.744±0.003  | 0.944±0.001      |

---

> > ### Comment · Reviewer_rWVP · 2023-11-21
> >
> > I thank the authors for their response. I am willing to keep my score and request the authors carefully revise the work, addressing all the feedback raised by the reviewers, especially on comparisons to prior works.

---

> > > ### Author Response · Authors · 2023-11-23
> > > **Response to Reviewer rWVP [2]**
> > >
> > > We sincerely thank the reviewer for keeping the positive grading and acknowledging the value of our work. We addressed the feedback in the latest revision and listed the main changes [here](https://openreview.net/forum?id=OGtnhKQJms&noteId=cY7dVjhdUH).

---

### Official Review · Reviewer_RoXZ · 2023-11-04

**Soundness:** 3 good
**Presentation:** 4 excellent
**Contribution:** 3 good
**Rating:** 8
**Confidence:** 3

**Summary:**

The authors study latent variable identifiability in the setting where there are multiple observations (views), each being a nonlinear mixture of a subset of latent components. Existing work identifies the shared latents for a given set of views $V$. This work generalizes the existing results, where the shared latents are identified for any subset of views $V_i \subseteq V$. Moreover, this is done simultaneously for all possible $V_i$ in a single training run. Many related identifiability results can be framed as a special case of the authors’ general framework.

**Strengths:**

This paper is written extremely well, and reads like a chapter from a textbook. The authors frequently refer to a running example and include “intuition” paragraphs to make it easier to understand their definitions and results. The loss that leads to identifiability (Eq. 3.1) is simple and intuitive, which makes me optimistic about the (eventual) practical applicability of this approach. The authors’ framework unifies many existing theoretical results in nonlinear ICA and causal representation learning, and also explains the empirical success of certain existing methods. This is a very strong paper that made me want to learn more about the related literature.

**Weaknesses:**

The main weakness of this work is its impracticality and lack of experimental results on realistic datasets. However, this is also acknowledged by the authors, and can also be said regarding most papers in this research area.

**Questions:**

I understand that it is more general to be able to simultaneously identify the content blocks for all possible $V_i$, but what practical benefit does this generality bring? Was this necessary in order to unify a wider range of existing identifiability results? If so, can you give a  specific example how this generality helped?

---

> ### Author Response · Authors · 2023-11-14
> **Response to Reviewer RoXZ**
>
> We thank the reviewer for the positive feedback, and the compliment about the significance of the paper. In the following, we list the **practical benefits** of simultaneously identifying all the blocks:
>
> - Our **intuitive motivation** is that we can measure the state of a system using multiple different sensors, each telling us something different. At the same time, we are interested in capturing the state of the system via high-level variables, and each block corresponds to a macro-variable (whose granularity depends on how fine-grained our sensors are). We believe this viewpoint will broaden the applicability of causal representation learning to more realistic and practical settings.
>
> + This is also **important for unifying results** from the disentangled representation literature, where the goal is to recover all factors of variation. With this, we can perfectly cover the theoretical results of (Locatello et al., 2020, Ahuja et al., 2022).
>
> - Additionally, we provide an interpretation for recent disentanglement work (Fumero et al., 2023) that **scales to real-world data** in settings where factors of variations are not independent.
>
> + Overall, we maintain that this setup is very flexible, covers important special cases, already explains some practical algorithms and we hope will lead to more.

---

> > ### Comment · Reviewer_RoXZ · 2023-11-20
> >
> > Thanks for pointing out that the generality is necessary in order to cover specific related works. I maintain my score recommending acceptance.

---

> > > ### Author Response · Authors · 2023-11-23
> > > **Response to Reviewer RoXZ [2]**
> > >
> > > We are glad we could answer the reviewer's question. We sincerely thank the reviewer for keeping the positive feedback and recommending acceptance.

---

### Author Response · Authors · 2023-11-14
**General response to all reviewers and the AC**

We thank the reviewers for their valuable feedback. We very much appreciate that they found this paper *“extremely well written and reads like a chapter of a textbook”* (`RoXZ`), *“communicates clearly motivations, formulation, technical details, and theoretical implications.”* (`QCXK`), and that they *“like how the paper is structured with an intuition behind definitions and theorems for readability”* (`BPer`). We also thank the reviewers for considering our work as *“a very strong paper that made me want to learn more about the related literature.”* (`RoXZ`), *“a neat unifying contribution”* (`rWVP`), a paper with *“thorough empirical evaluation”* (`QCXK`) and *“extensive literature review”* (`BPer`). Lastly, we are very grateful for the acknowledgement that our work is  *“technically well-executed and fits the conference”*(`rWVP`).

&nbsp;

Regarding the concern about the novelty and contribution of the paper, please allow us a quick and important clarification:

- **Our paper considers a novel setting**: multi-view casual representation learning under partial observability. This includes many special cases that have been previously studied, and we naturally build our results on top of prior work.

+ **Theorem 3.2: This is the closest connection with prior work because this theorem is a direct generalization thereof**. The proof directly generalizes Von Kugelgen et al., 2021, (and Daunhawer et al., 2023 that directly builds on the former) as they are limited to a pair of views. For theorem 3.1, one could imagine an alternate proof by induction over the number of views, where **Von Kugelgen et al., 2021, and Daunhawer et al., 2023 would be the base case**. We opted for a direct proof technique as the induction proof may have been perhaps more intuitive at a high level but was significantly longer. Additionally, we thought that a more familiar proof technique would be generally more accessible.

- **Theorem 3.8 invokes the result of Theorem 3.2 using a new proof** that builds on a ladder of results with weaker and weaker assumptions. We first show that identifiability is possible with a single model if the content factors ID are known, then we show how to learn content factors IDs from assumptions on their size, and finally how to identify the size.

+ **Identifiability Algebra corollaries**: here, we develop tools to extend proven identifiability results to novel settings, requiring an ad-hoc proof. The algebra tells what variables are identifiable simply by looking at graphical assumptions. It alone extends prior results such as Daunhawer et al., 2023, where they only identify the intersection of the views while the independent view-specific latents are also identifiable.

---

> ### Author Response · Authors · 2023-11-23
> **General Response [2]**
>
> ### **Change list**
>
> We sincerely thank all the reviewers for their constructive feedback and further references to improve this paper. Following their advice, we revised the manuscript (updates marked in green) and listed the major modifications here:
>
> - Add [1, 2] to Appendix B. Paragraph 1, clarifying that training deep generative models with a mixture prior could be considered as enforcing a soft alignment within the cluster, as pointed out by Reviewer `rWVP`.
> + Add **evaluation** on numerical simulations using **MCC** (Page 26, Table 3) and additional discussion (Appendix D.1 Paragraph 4). (`rWVP`)
> - Add a remark on the proof technique of Theorem 3.2 (Page 21), clarifying the **connection between our proof and prior work** [3, 4], as stated in the second point of the [general response](https://openreview.net/forum?id=OGtnhKQJms&noteId=qN34EnF5VV). (`QCXK`, `BPer`)
> + Discuss [5] in Appendix E. Paragraph 1 for possibilities to weaken the **invertibility assumption** using graphical constraints on the underlying causal structure. (`QCXK`)
>
> - Discuss recent work on **nonparametric** causal representation learning using **interventional** data ([1, 2, 6]) in Appendix E. Paragraph 5. (`BPer `)

---

> > ### Author Response · Authors · 2023-11-23
> > **References**
> >
> > - [1] Simon Buchholz, Goutham Rajendran, Elan Rosenfeld, Bryon Aragam, Bernhard Schölkopf, and Pradeep Ravikumar. Learning linear causal representations from interventions under general nonlinear mixing. arXivpreprint arXiv:2306.02235, 2023. 26, 31
> >
> > + [2] Julius von Kügelgen, Michel Besserve, Wendong Liang, Luigi Gresele, Armin Kekic, Elias Barein- boim, David M Blei, and Bernhard Schölkopf. Nonparametric identifiability of causal representations from unknown interventions. arXiv preprint arXiv:2306.00542, 2023. 1, 26, 31
> >
> > - [3] Julius von Kügelgen, Yash Sharma, Luigi Gresele, Wieland Brendel, Bernhard Schölkopf, Michel Besserve, and Francesco Locatello. Self-supervised learning with data augmentations provably isolates content from style. Advances in neural information processing systems, 34:16451–16467, 2021. 1,2,3,4,6,7,8,9,10,17,19,21,25,26,28,29,30
> >
> > + [4] Imant Daunhawer, Alice Bizeul, Emanuele Palumbo, Alexander Marx, and Julia E Vogt. Identifiability results for multimodal contrastive learning. In The Eleventh International Conference on Learning Representations, 2023. 1, 2, 3, 4, 6, 7, 9, 10, 17, 21, 25, 28, 29, 30
> >
> > - [5] Lingjing Kong, Biwei Huang, Feng Xie, Eric Xing, Yuejie Chi, and Kun Zhang. Identification of nonlinear latent hierarchical models. arXivpreprint arXiv:2306.07916, 2023. 30
> >
> > + [6] Wendong Liang, Armin Kekic, Julius von Kügelgen, Simon Buchholz, Michel Besserve, Luigi Gresele, and Bernhard Schölkopf. Causal component analysis. arXivpreprint arXiv:2305.17225, 2023. 31

---

### Meta-Review · Area_Chair_P8Mv · 2023-12-07

**Metareview:**

This paper studies the identifiability issue of causal representation learning in a multi-view setting where each view constitutes a nonlinear mixture of a subset of underlying latent variables.  The work extends several existing related theoretical results and is considered by the reviewer as an important step forward in causal representation.  Thorough empirical evaluation was carried on synthesis although there is a concern on the  lack of experimental results on realistic datasets.  The paper is well written with extensive literature review. It is technically well-executed and fits the conference.

**Justification For Why Not Higher Score:**

This line of work is conceptually interesting, and far from being practical yet.

**Justification For Why Not Lower Score:**

Strong reviewer support.

---

### Decision · Program_Chairs · 2024-01-16

Accept (spotlight)